# Genetically- and spatially-defined basolateral amygdala neurons control food consumption and social interaction

Hansol Lim [1], Yue Zhang[2], Christian Peters [1], Tobias Straub [3], Johanna Luise Mayer[1] & Rüdiger Klein [1] ✉

The basolateral amygdala (BLA) contains discrete neuronal circuits that integrate positive or negative emotional information and drive the appropriate innate and learned behaviors. Whether these circuits consist of genetically-identifiable and anatomically segregated neuron types, is poorly understood. Also, our understanding of the response patterns and behavioral spectra of genetically-identifiable BLA neurons is limited. Here, we classified 11 glutamatergic cell clusters in mouse BLA and found that several of them were anatomically segregated in lateral versus basal amygdala, and anterior versus posterior regions of the BLA. Two of these BLA subpopulations innately responded to valence-specific, whereas one responded to mixed - aversive and social - cues. Positive-valence BLA neurons promoted normal feeding, while mixed selectivity neurons promoted fear learning and social interactions. These findings enhance our understanding of cell type diversity and spatial organization of the BLA and the role of distinct BLA populations in representing valence-specific and mixed stimuli.

The basolateral complex of the amygdala (BLA) processes and encodes the emotional valence of salient cues and controls the appropriate behavioral output. In the past, behavioral studies were inspired by BLA lesions which caused profound behavioral disturbances, typically focusing on motivated behavior, including aversive[1,2], appetitive[3–5], and instrumental conditioning paradigms[6–9], as well as addictive[10,11] and anxiety behaviors[12–15]. The BLA has also been implicated in social behavior in humans and rodents[13,16,17]. Social behavior is complex and can be rewarding, but also upsetting. It activates the brain's reward circuitry[18], but can also trigger avoidance and aggression[19–21]. Hence, social interactions can have positive and negative valence[22]. Optogenetic inactivation of the BLA facilitates social interactions[23], suggesting that the BLA specifically mediates aversive aspects of social interaction. Consistent with such a conclusion, the BLA projection to ventral hippocampus (vHPC) mediates social and anxiety-related

behaviors[24]. More recently, it was shown that the BLA encodes social exploration behavior in a valence-independent manner by two functionally anticorrelated ensembles, consistent with a multiplexed representation of valence in the BLA[25].

Somewhat surprisingly, the BLA has not been implicated in the normal consumption of food. Rather, published work indicates that BLA principal neurons (PN) promote reward (sucrose) consumption in an instrumental goal-directed action performance test[26], and that BLA projections to Nucleus Accumbens (NAc) enhanced instrumental food consumption in a chronic stress paradigm, but not free-feeding behavior[27]. Moreover, BLA activity decreased the likelihood of food consumption in a risky environment[28], suggesting that the BLA regulates risk behavior when animals forage for food. The fact that some BLA neurons increase firing in response to food[29,30], suggests the presence of distinct BLA PN populations that contribute to the control of normal feeding.

[1]Department Molecules – Signaling – Development, Max Planck Institute for Biological Intelligence, Martinsried, Germany. [2]Department Synapses – Circuits - Plasticity, Max Planck Institute for Biological Intelligence, Martinsried, Germany. [3]Biomedical Center Core Facility Bioinformatics, LMU, Munich, Germany. ✉e-mail: ruediger.klein@bi.mpg.de

Current models posit that aversive and rewarding stimuli may involve discrete neural circuits to elicit responses ranging from defensive to appetitive behaviors[31,32]. The distinction of these neural circuits is, however, incompletely understood and many questions remain unanswered: Are the BLA neurons that encode positive or negative values genetically distinct and do they occupy discrete spatial locations within the BLA? Do individual BLA neurons innately represent one type of stimuli, or can they represent mixed selectivity? Currently, there is evidence for all of these scenarios. Genetically distinct BLA neurons controlling valence-specific motivated behavior have been previously described[33,34]. Conversely, distinct appetitive and aversive networks in BLA were described based on their segregated projections to NAc and central amygdala (CeA)[35,36]. Such output specificity may be determined during development by genetically encoded guidance cues, but this remains to be demonstrated[37]. Single-cell recordings have shown that genetically defined BLA neurons are typically quite heterogeneous with fractions of cells responding to positive and negative cues[30,34,38]. To what extent these variable responses to multiple stimuli reflect the cell type or the result of changes in the internal state of the animals, including anxiety, arousal, sensory processing, or previous experience remains to be determined[39–41].

One experimental approach that could provide insights into this diversity of BLA responses is single-cell transcriptomics (scRNAseq) which enables the identification of cell types based on similar gene-expression properties[42]. Coupling transcriptomic cell typing with their spatial organization, input-output pathways, and functional analyses has the potential to reveal the organizational framework of the BLA. A previous scRNAseq analysis of excitatory neurons of the BLA revealed two main principles: First, the anatomical division of the BLA into lateral (LA) and basal amygdala (BA) was recapitulated in a discrete separation of transcriptomic cell types arguing that excitatory neurons of the LA and BA should be considered distinct[43]. Second, within LA and BA, the analysis revealed anterior-posterior continuous gene-expression differences instead of discrete cell types. However, this organization was inconsistent with previous anatomical parcellations of the BLA[44–46] and no functional analysis was included.

Another recent scRNAseq study of the whole amygdala complex[47], found genetic differences among amygdala nuclei and between two types of glutamatergic neurons, *Slc17a7* (*VGLUT1*) and *Slc17a6* (*VGLUT2*). Also, after fear conditioning several cell types with higher immediate early gene (IEG) expression were identified for putative roles in aversive memory[47]. However, the association of transcriptomic cell clusters with subregions of the BLA was less clear due to a lack of spatial resolution and again no functional analysis was included.

In this work, we provide a full description of the genetic diversity of glutamatergic BLA neurons and their spatial distribution. We characterized a total of 11 cell clusters and demonstrated that they were distributed in distinct spatial BLA subregions. Unsupervised approaches using dimensionality reduction and cross-correlation of meta datasets suggest that distinct cell clusters are present within LA and BA subregions. We selected three genetically defined neuron populations each representing two or three transcriptional clusters, for subsequent functional analysis. We characterized a positive-valence subpopulation that increased activity in response to the presence of food and promoted feeding. A second population showed mixed selectivity by responding to aversive and social cues and not only promoted defensive but also social behavior. A third population followed the classical BLA model of an aversive population by not responding to food and social cues and promoting aversive conditioning. These findings enhance our understanding of cell type diversity and spatial organization of the BLA and the role of distinct BLA PN populations in innate food consumption and social behavior.

## Results

### Single-nuclei RNAseq identifies molecularly defined glutamatergic BLA cell types

We characterized transcriptomic cell types using single-nuclei RNA sequencing (snRNAseq) from the adult BLA (4 male brains with both hemispheres), spanning 2.4 mm in anterior-posterior direction (Fig. 1A). To spatially annotate the cell clusters, we used a regional parcellation method as published previously[46] (Fig. 1B). Our initial analysis of the BLA transcriptomic dataset identified seven transcriptomic cell types, including 3,278 non-neuronal and 4,544 neuronal cells (Supplementary Fig. 1A, B). We re-clustered the neuronal cells only and separated them into GABAergic and glutamatergic neurons using specific markers for inhibitory (*Gad1, Gad2* and *Slc32a1*) and excitatory neurons (*Slc17a7*) (Fig. 1C, Supplementary Fig. 1C). Separate re-clustering of GABAergic neurons revealed 10 clusters, including intercalated cells (ITC) and amygdalostriatal area cells (marked by *Foxp2* and *Rarb*, respectively) and a cluster marked by *Tshz2* and *Rmst* which appeared to be equally related to ITC and BLA GABAergic interneurons (Supplementary Fig. 1D, E). The latter neurons were separated into two populations marked by *Reln* and *Calb1* (Supplementary Fig. 1E). The *Calb1* population included *Calb1, Sst* and *Htr2a* clusters, while the *Reln* population contained *Lamp5, Ndnf*, and *Cck* clusters (Supplementary Fig. 1E), showing distinct transcriptomes and correlations with each other (Supplementary Fig. 1F). These findings agreed well with recent scRNA-seq studies from cortex and BLA[47–49].

Separate re-clustering of glutamatergic BLA cells identified 12 clusters including one cluster (cluster 10) from posterior medial amygdala (MEAp) marked by *Esr1* and *Pde11a*[50] (Fig. 1D, E). Compared to GABAergic clusters, glutamatergic clusters shared many of their top 5 marker genes. For example, all top 5 genes for cluster 2 (cl2) were also expressed in cl6. *Rspo2* was expressed in cl1 and 11, and *Rorb* in cl7 and cl8 (Fig. 1D). To identify distinctive marker combinations for glutamatergic clusters, we first calculated the top marker genes for each cluster using an AUC (area under the curve) threshold of 0.5 and a minimum observation percentage of 20 (see methods). We then chose genes expressed exclusively in one or two clusters. Through this analysis, we identified a minimum of three representative genes for each cluster (Fig. 1F). We then inspected the spatial expression of each marker in the Allen brain atlas and selected 10 genes that appeared to be expressed in subregions of the BLA (Supplementary Fig. 1G). Marker genes that were expressed homogeneously throughout the BLA or showed high expression in nearby brain regions were not selected (Supplementary Fig. 1H). Some of the selected genes were strongly enriched in one cluster, including *Sema5a* (cl5), and *Grik1* (cl12), while others were enriched in two or more clusters, including *Rorb* (cl7,8), *Otof* (cl2,3,6) and *Lypd1* (cl2,8,9) (Fig. 1F). By this analysis, every cluster of glutamatergic BLA neurons could be represented by a combination of one to four marker genes (Fig. 1E, F). This approach provided a combinatorial set of marker genes for spatial mapping of molecularly defined glutamatergic BLA cell types.

### Spatial organization of glutamatergic BLA clusters

For spatial mapping of glutamatergic BLA clusters, we performed sequential multi-plexed fluorescent in situ hybridization (smFISH) from anterior to posterior whole BLA coronal sections (Supplementary Fig. 2-3). We used four coronal sections−designated "anterior", "anterior-middle", "posterior-middle" and "posterior"−to divide the BLA into eight subregions from anterior to posterior (aLA, pLA, ppLA, amBA, alBA, acBA, pBA, ppBA) according to published methods[46] (Fig. 2A−D). The numbers of cells positive for each marker gene were counted and the fractions of positive cells in each subregion were analyzed (Fig. 2E, see Methods). Briefly, we set a threshold on the fluorescence signal of each marker gene to determine whether a cell was positive or negative for a particular marker and then calculated the percentages of positive cells for each gene within each subregion.

Since some cells were positive for multiple markers, the sum of cell fractions per subregion exceeded 100%. This analysis also allowed us to compare cell abundance across the different subregions. The results indicated that the clearest distinction was between LA and BA. For example, *Rorb*-positive cells were enriched in LA and less frequent in BA subregions (Fig. 2A3–D3, E). *Etv1* and *Rspo2*-positive cells showed

the opposite pattern (low in LA, high in BA), which was most obvious in anterior and anterior-middle sections (Fig. 2A1, 2-B1, 2). We also observed that cell distributions varied in the A-P axis. For example, *Cdh13* expression was nearly absent in aLA and enriched in pLA/ppLA (Fig. 2A2–D2), whereas *Rorb* was enriched in aLA and less so in ppLA (Fig. 2A3–D3). Some patterns were more complex: *Lypd1*-positive cells

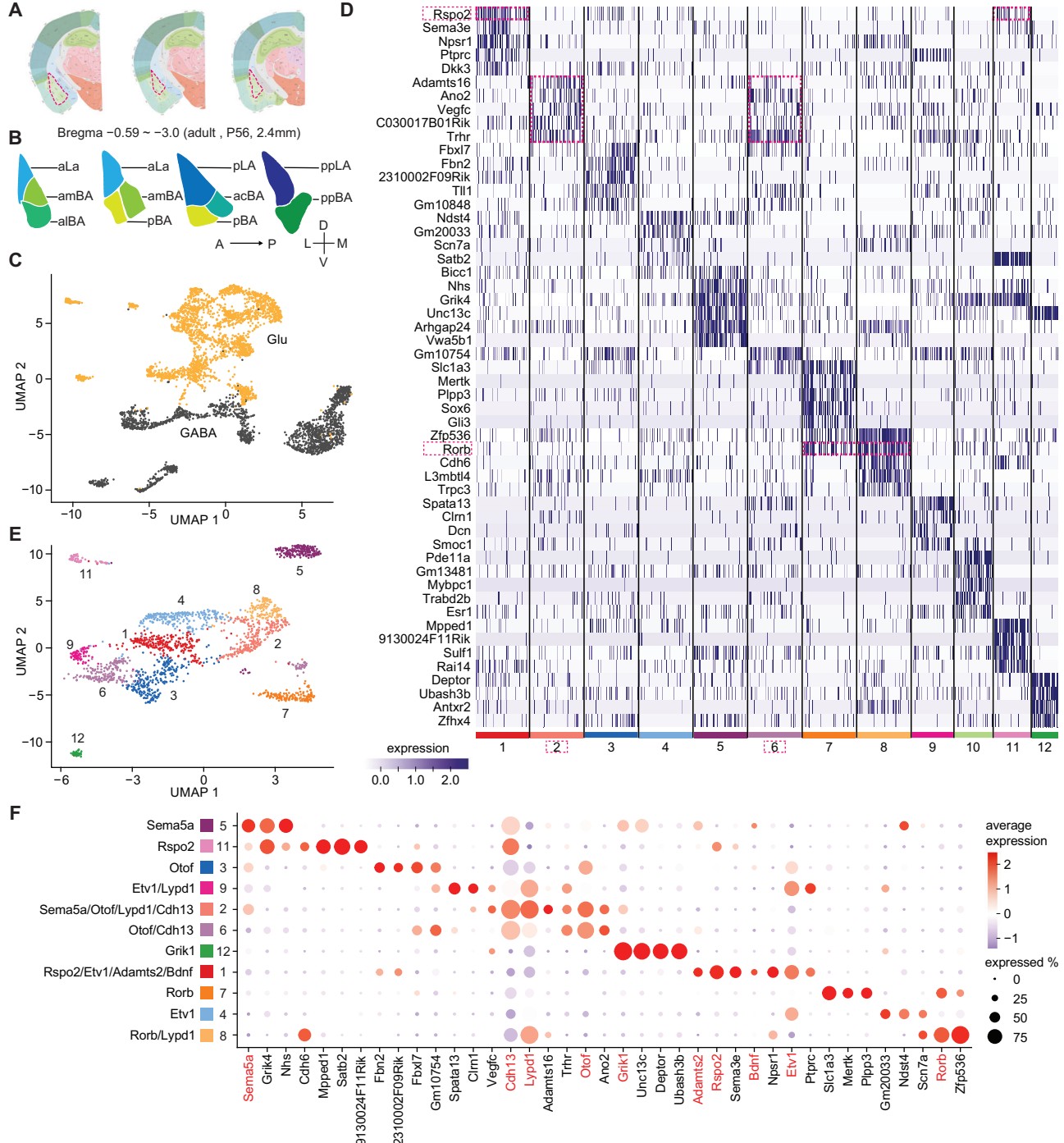

**Fig. 1 | Single-nuclei transcriptomic characterization of adult BLA neuron types. A** Schemes showing the sampled BLA regions highlighted with a triangle. The anterior-posterior extent of the samples ranged from bregma 0.59 to −3.0 covering around 2.4 mm. **B** Schemes showing regional parcellation of the BLA along the anterior-posterior axis (adapted from ref. 46). acBA, anterior-caudal BA; aLA, anterior LA; alBA, anterior-lateral BA; amBA, anterior-medial BA; pBA, posterior BA; pLA, posterior LA; ppBA, posterior-posterior BA; ppLA, posterior-posterior LA. **C** UMAP of BLA neurons (*n* = 4544) with cells classified as GABAergic

(GABA, *n* = 2033, black) and glutamatergic (Glu, *n* = 2511, orange), respectively. **D** Heatmap of the top 5 marker genes in each cluster of glutamatergic neurons. **E** UMAP of glutamatergic neuron clusters after separate dimension reduction and clustering. Cell type color palette reflects the one shown in (**F**). **F** Molecular signatures of glutamatergic clusters in dot plot visualization of average gene expression of selected candidate genes. Genes highlighted in red were selected as ten key markers; percentage of cells expressing the selected marker is indicated by circle size and average gene expression level by color scale.

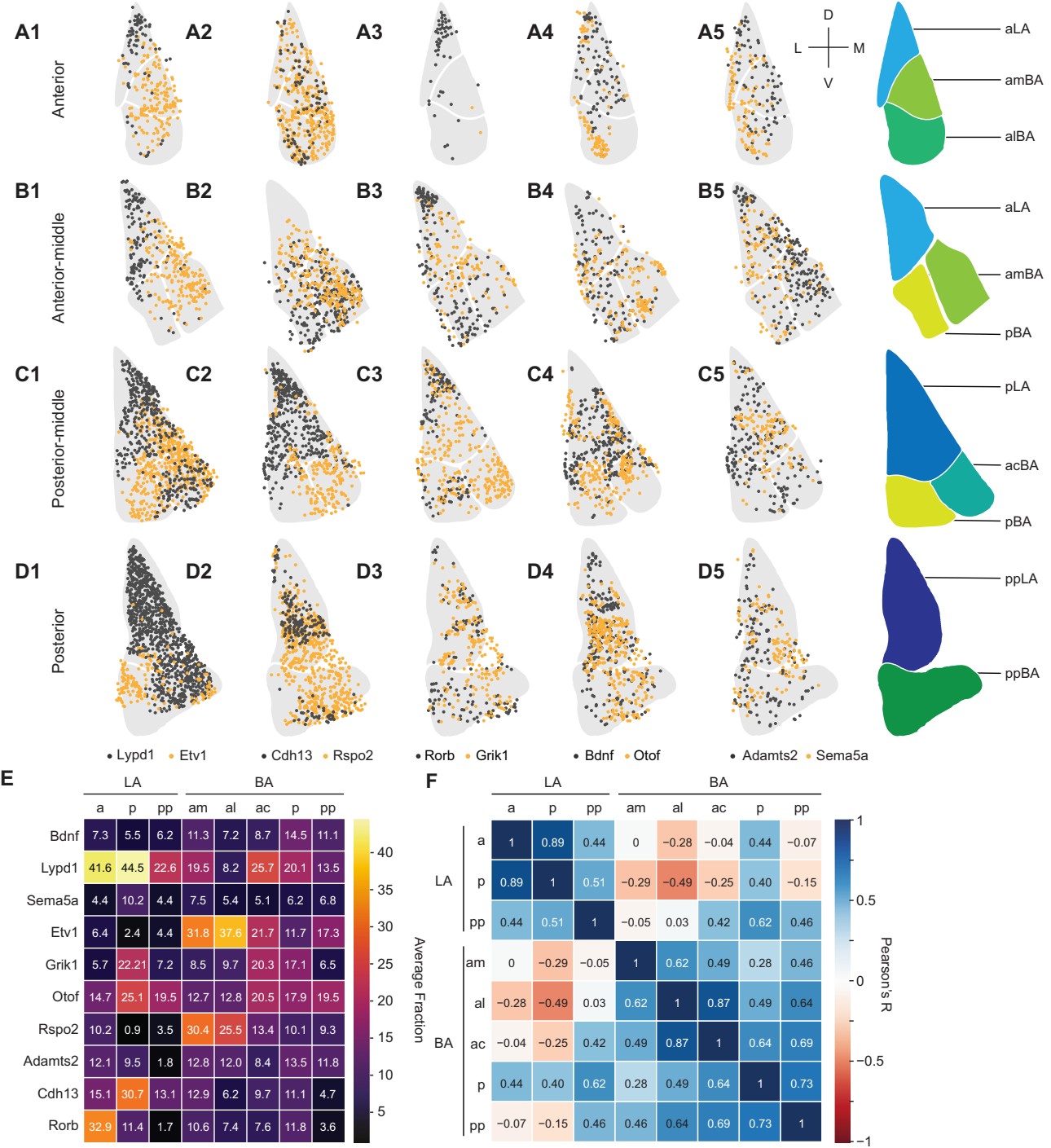

**Fig. 2 | Distribution of cells positive for selected marker genes in BLA subregions. A1–D5** Distribution of cells within BLA subregions along the anterior-posterior axis. Examples of five pairs of marker genes colored in black and yellow are shown from 1 to 5 while anterior to posterior sections are ordered from (**A**) to (**D**). **E** Quantification of the distribution of cells positive for a specific marker gene within BLA subregions. Heatmap indicating large fractions of cells in yellow and small fractions in dark purple (Average fraction size in percent is indicated in each tile). **F** Pearson correlation of averages on the percentage of cells expressing each gene in eight subregions; colors indicate Pearson's R.

were enriched in the whole LA and several parts of the BA (Fig. 2A–E). *Grik1*-positive cells were scarce in most anterior sections and enriched in posterior parts of LA and BA (Fig. 2A3–D3, E).

Next, we asked if the pattern of cell distribution based on the expression of the ten marker genes would be sufficient to delineate subregions of BLA. Pearson correlation revealed that LA and BA separated clearly, which could be further subdivided in the A-P axis, separating aLA from pLA/ppLA, and amBA/alBA from the more

posterior acBA/pBA/ppBA (Fig. 2F). These results indicate that cells with similar expression profiles were distributed in distinct patterns in BLA subregions.

**The combination of marker genes predicts the spatial localization in the BLA**

During the above analysis, we noticed that the distribution of cells did not always follow the boundaries of the subregions, raising the

possibility that subregions could be further subdivided or arranged differently. For example, *Otof*-positive cells were highly concentrated at the lateral edge of the aIBA and rather scarcely present in the rest of the aIBA (Fig. 2 A4) and *Cdh13*-positive cells were enriched at the tip of pLA (Fig. 2C2). Since every cell was characterized by a unique combination of marker genes and its unique space within the BLA, we next asked in an unsupervised way, whether cells with a similar combination of marker genes would localize to a similar subregion of the BLA. For this, we used principal component analysis (PCA) to examine the variation of cells with their unique marker gene combinations and reconstructed the spatial localization of each PC back into the BLA. As the eigen-images from the top 4 PCs explained on average $80.8 \pm 5.16\%$ of variance in each sample, four PCs were enough to represent the variance. The results indicated that the PC associated with the largest variation in gene expression corresponded to differences between LA and BA (Supplementary Fig. 4). PCA also revealed that the markers *Etv1*, *Rspo2* and Lypd1 had large loadings in the top 4 PCs that demarcate the boundary between LA and BA. For example, *Etv1* and/or *Rspo2*-positive cells contributed most to BA-specific PCs (Supplementary Fig. 4A1, B2, C2, D2), while *Lypd1*-positive cells contributed most to LA-specific PCs (Supplementary Fig. 4B3, C3, D3, D4). Also, *Cdh13*-positive cells contributed to LA-specific PCs (Supplementary Fig. 4B1, C3, D4). In summary, our findings suggest that cells with a similar combination of marker genes localized to similar subregions of the BLA. Moreover, the expression patterns of *Etv1, Rspo2* and *Lypd1*, parcellated the BLA into its LA and BA subdomains. These results show that genetically marked cell populations distribute in distinct BLA subregions when analyzed in an unsupervised way.

To enhance the reliability of our transcriptomic and spatial expression analyses, we conducted a side-by-side comparison on the expression correlation of marker genes from snRNAseq (Fig. 1) and smFISH data (Fig. 2), respectively. In terms of transcriptional profiles, *Etv1, Bdnf, Adamts2,* and *Rspo2* exhibited a strong correlation and clearly separated from the other genes (Supplementary Fig. 5A). An analogous analysis using smFISH data revealed a similar trend: *Etv1, Bdnf, Adamts2,* and *Rspo2* were tightly correlated and predominantly found in the anterior BA (Supplementary Fig. 5B). Notably, the correlation trends remained consistent throughout. *Otof, Lypd1,* and *Cdh13* consistently showed close correlations and were primarily expressed in the posterior LA, while Sema5a and Grik1 were closely correlated and appeared predominantly in the posterior BA (Supplementary Fig. 5A, B).

## Transcriptional clusters with distinct expression patterns in BLA subregions

Until now, we used the data from smFISH to assign each cell to a specific BLA subregion. Next, we assigned each transcriptomic cell cluster to a space in the BLA in an unsupervised way using Pearson correlation analysis. Since every cell belonged to a transcriptomic cell cluster and, in addition, was characterized by the smFISH read counts of ten marker genes, we could calculate the scaled sums of expression of the ten marker genes for each cluster (Supplementary Fig. 6A). For example, cluster 2 was characterized by high expression of *Sema5a, Otof, Lypd1* and *Cdh13* (0.96-1.0) and lower expression of *Etv1* (0.63) and *Rspo2* (0.47).

In the smFISH data, every cell was characterized by the normalized expression of ten marker genes. We therefore correlated (Pearson) the smFISH marker expression pattern (Supplementary Fig. 6A) to each transcriptomic cluster marker expression pattern and assigned each smFISH cell to one of the 11 clusters according to the highest correlation coefficient (Supplementary Fig. 6B). For example, one smFISH cell (Cell ID # 3234) with high *Etv1* expression had highest correlation coefficient with cl4 (R: 0.46) and was therefore assigned as cl4. Another smFISH cell (cell ID # 1936) with high *Rspo2* expression showed higher correlation coefficient to cl11 (R:0.46), while a cell (cell ID # 5065) with high Lypd1

expression showed higher correlation coefficient to cl2 (R: 0.71) (Supplementary Fig. 6B). After the assignment of all smFISH cells to individual transcriptional clusters, all cells were reconstructed into BLA space by using the spatial coordinates from smFISH (Supplementary Fig. 6C). For example, cl1, and cl11 were enriched in anterior BA, cl4 more posterior BA (Fig. 3A). These three clusters were characterized by high expression *Rspo2* and *Etv1*. Instead, cl2 and cl8 were enriched in pLA or ppLA (Fig. 3A) and were characterized by high expression of *Lypd1*. cl7 was enriched in anterior LA and was characterized by high expression of *Rorb*. By this analysis, transcriptomic clusters could be annotated to BLA subregions. (Fig. 3B).

Using the average expression of the ten marker genes for each cluster we analyzed the correlation between clusters and generated a dendrogram (Fig. 3C). We annotated the tree with BLA subregions from the above analysis. Those clusters localized to LA correlated better compared to BA clusters. In each branch of the tree, clusters are separated according to their A-P axis. (Fig. 3C). In summary, these results from the unsupervised analysis suggest that transcriptional cell clusters of glutamatergic neurons distribute in distinct BLA subregions. In the D-V axis, there is a clear separation of cell clusters, while in the A-P axis, we find both segregated clusters, but also gradual expression changes.

## Genetically and spatially defined neurons show different response properties

We selected three genetic markers for functional analysis. Lypd1, the marker with the highest expression in LA and additional regions in BA, targeting three transcriptomic clusters (cl2,8,9); Etv1, the marker with scarce expression in LA, complementary pattern with Lypd1 in BA subregions, targeting three clusters (cl1,4,9); Rspo2, scarce expression in LA, restricted pattern in anterior BA, partially colocalizing with Etv1, targeting two clusters (cl1,10). Among these markers, only Rspo2 had previously been analyzed functionally and will serve as a reference for comparison[33]. A comparative mRNA expression analysis revealed that Lypd1-expressing cells showed little colocalization with Etv1- or Rspo2-expressing cells (typically less than 20% colocalization in BLA, Supplementary Fig. 7A–C). Etv1- and Rspo2-expressing cells colocalized strongly in anterior sections (55%) and much less in posterior sections (30%) (Supplementary Fig. 7C). The fraction of Lypd1-expressing cells increased from anterior to posterior, while those of Rspo2- and Etv1-expressing cells decreased from anterior to posterior.

To analyze the intrinsic physiological properties of the neurons marked with the three selected genes, we used the respective Cre lines, Lypd1-Cre, Etv1-CreER, and Rspo2-Cre, and validated Cre expression in comparison to the endogenous markers (Supplementary Fig. 8A–E and Supplementary Fig. 9). Additionally, our snRNAseq data showed that BLA[Etv1] and BLA[Lypd1] neurons were mainly localized in glutamatergic, rather than GABAergic cells (Supplementary Fig. 8F), and quantification of Lypd1-Cre or Etv1-CreER; tdTomato expression with *Slc17a7* (*VGLUT1*) showed more than 80% co-labeling (Supplementary Fig. 8G, H). With this Cre-line validation, we performed ex vivo electrophysiology in brain slices. Whole-cell current-clamp recordings revealed significant differences in the membrane potentials with BLA[Lypd1] neurons showing the most negative and BLA[Rspo2] neurons the least negative (Fig. 4A), suggesting that BLA[Lypd1] neurons may require more excitatory inputs to fire than the other two. Basic firing rates did not differ greatly between cells (Supplementary Fig. 10A); however, spontaneous excitatory postsynaptic currents (sEPSC) had a lower amplitude and spontaneous inhibitory postsynaptic currents (sIPSC) had lower frequency in BLA[Lypd1] than BLA[Rspo2] cells (Supplementary Fig. 10B), suggesting that BLA[Lypd1] cells express fewer glutamate receptors and receive fewer inhibitory inputs.

Since part of our analysis involved appetitive behavior, which is known to be controlled by BLA neurons[31,33], we asked if overnight fasting would modify neuronal activities. Current-clamp recordings

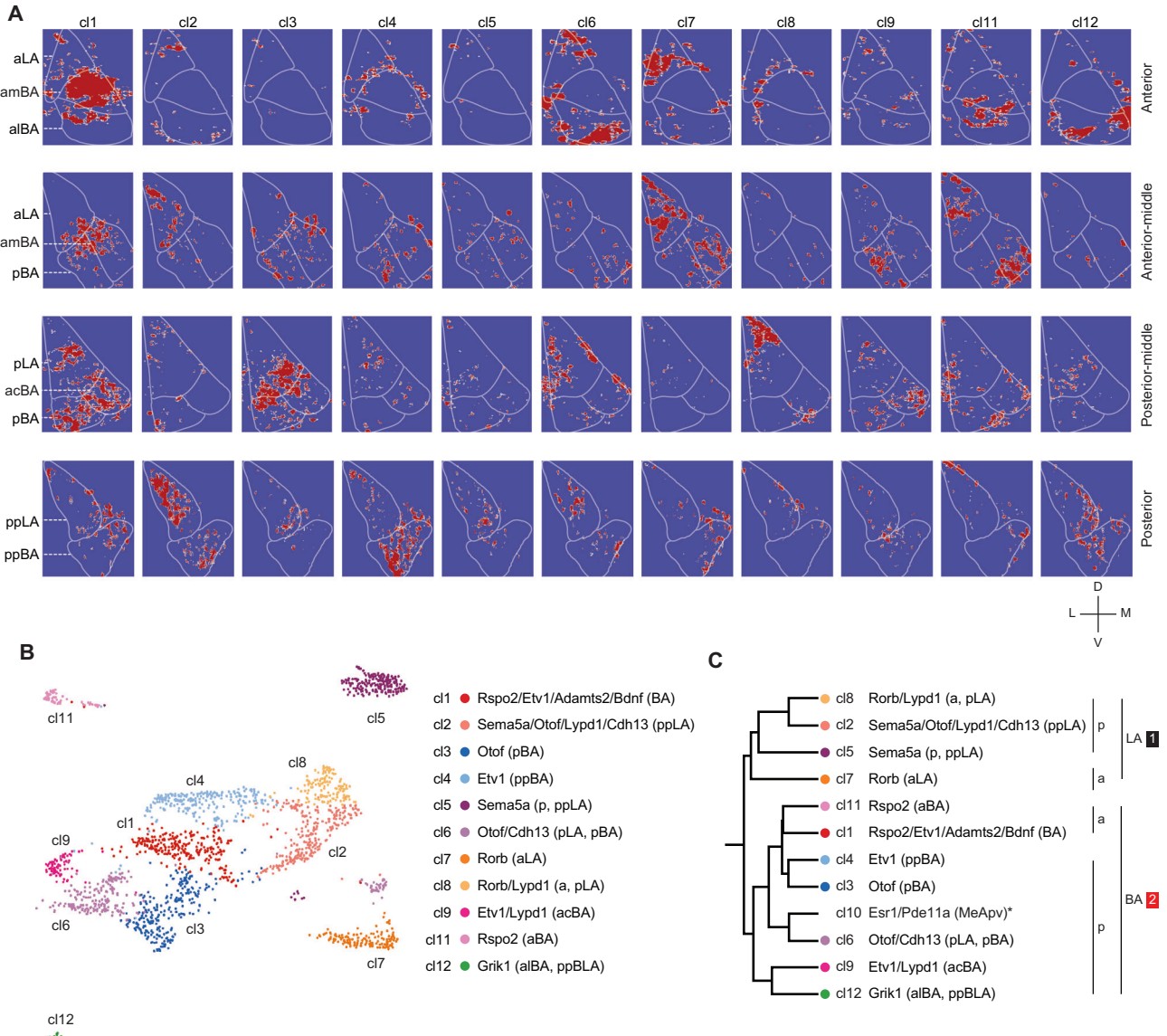

**Fig. 3 | Spatial expression of transcriptional clusters. A** Mapping of snRNA-Seq clusters (11 clusters) to smFISH signals and corresponding locations in the BLA; Panels from top to bottom indicate anterior to posterior sections and colors represent binary expression (Red= expression, Blue = no expression, each cell ID is created after normalization within a radius of 50 μm). **B** Final annotation of snRNA clusters with respect to expression of markers genes and their distribution in BLA subregions. **C** Dendrogram of snRNA clusters based on hierarchical clustering of aggregated mRNA expression (Subregions were categorized into large two categories either LA or BA (a, anterior and p, posterior)).

showed that the firing rates of $BLA^{Lypd1}$, but not $BLA^{Etv1}$ neurons, increased in fasted mice and their membrane potentials depolarized (Fig. 4B–F). Since the depolarization of the membrane potential brings the neuron closer to the threshold for firing, this contributes to an increase in the firing rate of the fasted $BLA^{Lypd1}$ neuron. This suggests that $BLA^{Lypd1}$ neurons were more excitable after fasting. Recordings of excitatory and inhibitory neurotransmission revealed increased frequencies of sEPSC and sIPSC in $BLA^{Lypd1}$ neurons after fasting (Fig. 4G–I). Additionally, the decay time for sEPSC decreased in $BLA^{Lypd1}$ neurons after fasting, suggesting changes in the kinetics of the excitatory receptors (Fig. 4J). Other electrophysiological parameters measured in $BLA^{Lypd1}$ neurons did not change after fasting (Supplementary Fig. 10C). Together these results suggest that the physiological properties of $BLA^{Lypd1}$ neurons change during periods of energy deficits.

To understand how these BLA neuron populations modulate appetitive and defensive behaviors, we performed single-cell-resolution in vivo calcium imaging in freely moving mice. A graded-index (GRIN) lens was implanted above the BLA in the respective Cre lines previously injected with an AAV expressing a Cre-dependent GCaMP6f calcium indicator (Supplementary Fig. 10F). Calcium activity was monitored with a head-mounted miniaturized microscope in a free feeding assay[51] (Fig. 4K). We quantified mouse feeding behavior according to their approach behavior towards food rather than food consumption because, in previous work on central amygdala neurons, the presence of food correlated better with neuron activity than food consumption[52]. Also, manual scoring by different observers agreed that within a distance of 5 cm around the food dish, the mouse was very likely to consume food (in 9/10 cases). To visually inspect the correlation between the neural activities and the distance to food, we plotted the firing rate inferred from the calcium traces (see Methods for details) with the behavior trace (Fig. 4L). We observed many neurons with substantially high firing rates in specific areas. In some of these neurons, these high firing rate areas partially overlapped with the

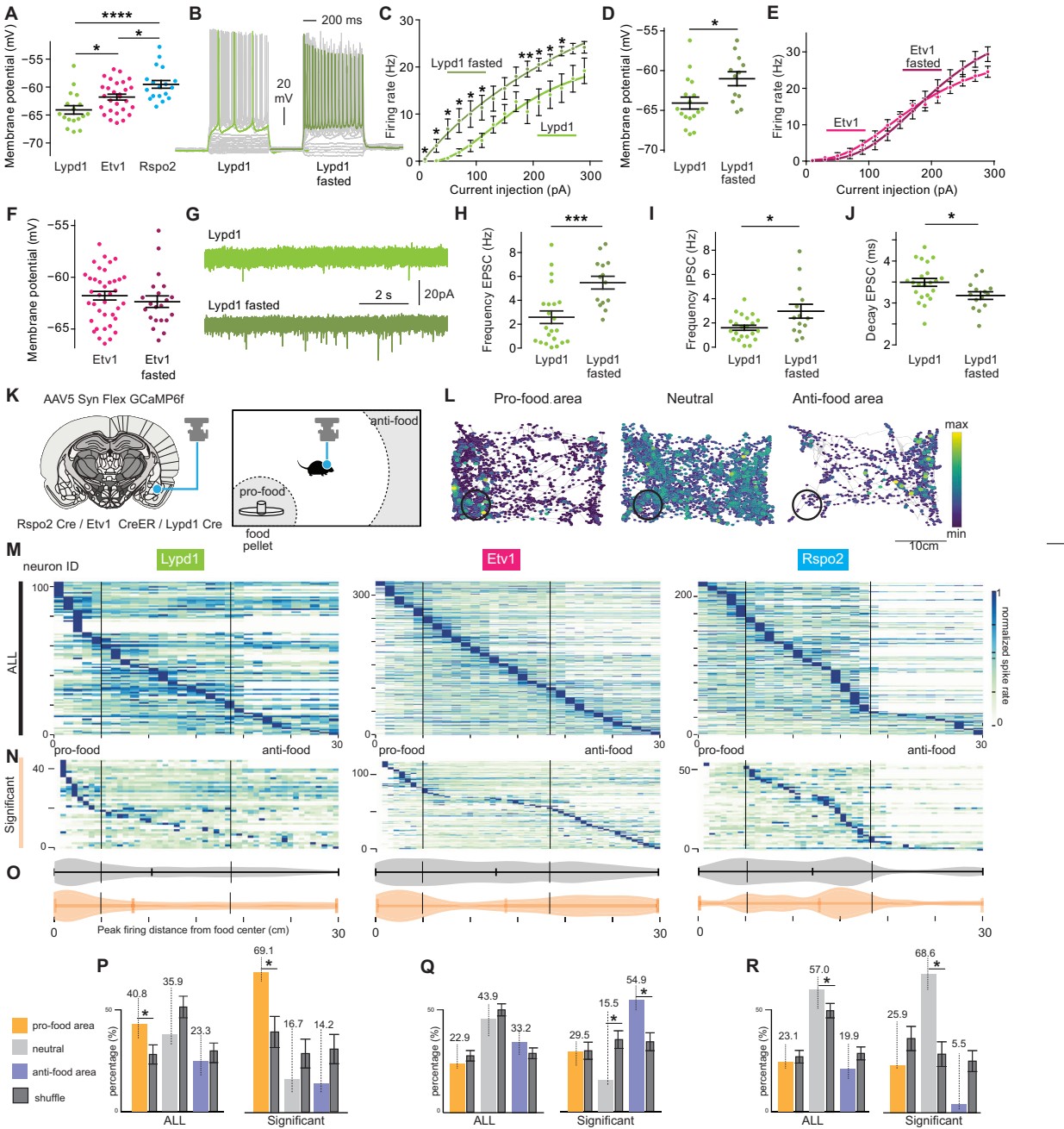

**Fig. 4 | Genetically and spatially defined neurons show different feeding-related activities in vivo _and_ in vitro. A** Membrane potentials of BLA sub-populations (One-way ANOVA, Bonferroni corrected) (*n* = 2 mice per group and cell number: Lypd1 = 17, Etv1 = 28, and Rspo2 = 18). Adjusted P: Etv1 vs Rspo2: 0.0321, vs Lypd1: 0.0356, Rspo2 vs Lypd1: <0.0001. **B** Whole-cell current-clamp recordings of BLA$^{Lypd1}$ neurons from fed and fasted animals showed all current steps applied (gray) with a highlighted recording (green) at 230 pA. Firing rates (Hz) after injecting different current steps in BLA$^{Lypd1}$ (**C**) and BLA$^{Etv1}$ neurons (**E**). Two-way ANOVA Mixed-effects analysis: $F(1, 29) = 6.997$, $P = 0.0130$ for Lypd1, marked in corresponding steps. Lypd1 fed: *n* = 17, Lypd1 fasted: *n* = 12. Etv1 group: Fed vs Fasted $P = 0.8375$, Etv1 fed = 38, Etv1 fasted: *n* = 21 cells. Membrane potentials in Lypd1 (**D**), Etv1 neurons (**F**). Unpaired *t* test (*p* = 0.0135). Lypd1 fed: *n* = 17, Lypd1 fasted: *n* = 12. Etv1 fed = 38, Etv1 fasted: *n* = 21cells (*P* = 0.4680). **G** sEPSC recordings in Lypd1 neurons of fed and fasted animals Quantification of sEPSC frequency (**H**), sIPSC frequency (**I**), and sEPSC decay (**J**) in Lypd1 neurons of fed and fasted animals.

Unpaired *t* test; *p* = 0.0008 (**H**), 0.0135 (**I**), 0.0299 (**J**), Lypd1 fed: *n* = 22, Lypd1 fasted: *n* = 14 cells. **K** Schematic explanation of GRIN lens above GCaMP6f-expressing BLA neurons and free-feeding assay; modified from Allen Mouse Brain Atlas, mouse.brain-map.org (**L**) Response map of 3 example neurons in three categories with the distance to the food chamber. They averaged firing rate heat-maps in all (**M**) and in 'significant' neurons, significantly correlated to food distance determined with a permutation test (**N**). $N$ = 103, 328, and 221 cells were recorded from 12, 9, and 7 of BLA$^{Lypd1}$, BLA$^{Etv1}$, and BLA$^{Rspo2}$ mice, respectively. **O** Violin plots of the peak firing distance of neurons (significant (orange) and all neurons (gray)). **P**–**R** The percentages of neurons with peak firing rates in pro-food, neutral, and anti-food areas indicate significant differences (two-tailed test, Shuffle data with mean values ± SD). Significant cell: Lypd1; *n* = 29,7, and 6, Etv1; 43, 18, and 57, Rspo2; 14, 37, and 3 from the pro-, neutral, and anti-food, respectively. Data are presented as mean values ± SEM (**A–J**) or SD (**P–R**). *$p$ < 0.05, **$p$ < 0.001, ****$p$ < 0.0001, with all *t* tests being two-tailed.

location of the food container, while we observed the opposite in other neurons (Fig. 4L). To quantitatively assess the relationship between firing rate and the distance to food, we used spike detection to deconvolve calcium traces, divided the distance to food into 31 bins, and computed the average firing rate at each distance bin for each neuron. The area in which we observed feeding behavior, was within a 5 cm radius around the food container and was termed the pro-food area. The area in which no approach towards food was observed, was outside a 17 cm radius around the food container and was termed the anti-food area. The area in between was termed neutral area (Fig. 4L–N). For each neuron, we determined the peak firing rate, which was then used to sort the neuron into pro-food, neutral, or anti-food areas (Fig. 4M, N). We also performed a permutation test to identify the neurons whose activity was significantly correlated with the distance to food (see method for details). In brief, we pooled all neurons from the 3 populations and shuffled the spike train for each neuron with randomly selected N neurons. This creates a null distribution which hypothesizes the variance in the data can be explained by population-irrelevant factors. Using this empirical null distribution, we constructed a non-parametric permutation test to analyze how significant the variance is to reject the null hypothesis without making other assumptions to avoid any potential analysis biases (for example, the normality assumption which expects the values to be centered at 50%). Therefore, we determined if the neuron was considered significantly tuned to distance to food (termed "significant neurons"), by calculating if the maximum average firing rate of the distance distribution of a neuron was higher than 95% of the null distribution (Fig. 4N, O). This analysis revealed that the food-distance sorting pattern of the significant neurons was consistent with the pattern of all neurons (Fig. 4M–O).

The quantification of all neurons revealed that the largest fraction of active neurons in the BLA$^{Lypd1}$ population was in the pro-food area (40.8%) (Fig. 4P). This percentage value was statistically significant when we computed the null percentage distribution of all pooled neurons from the 3 populations (shuffled data), randomly selected N neurons (N equals the number of neurons in the tested population), and compared the percentage of recorded data with the shuffled data (Fig. 4P, left). Consistently, the percentage of BLA$^{Lypd1}$ significant neurons in the pro-food area was significantly higher than the chance level in comparison with the shuffled data of significant neurons from all three populations (Fig. 4P, right). This was in contrast to the BLA$^{Etv1}$ population, where only the anti-food fraction in the significant neurons was larger than the chance level (Fig. 4Q, right). In the case of BLA$^{Rspo2}$ neurons, the fractions of active cells in the neutral area across both all and significant neurons were larger than the chance level (Fig. 4R). The averages of food consumption for recorded mice were similar across Cre lines (Supplementary Fig. 10H). These findings confirmed that the representation of neuronal activities according to distance-based food preference is statistically reliable and further revealed that BLA$^{Lypd1}$ neurons were activated during fasting and food approach behavior.

## BLA$^{Etv1}$ neurons are activated by innate fear stimuli

In previous work, BLA$^{Rspo2}$ neurons were activated by electric foot-shocks during contextual fear conditioning (CFC)[30]. We therefore asked, what fractions of BLA$^{Lypd1}$ and BLA$^{Etv1}$ neurons were activated by these negative valence stimuli. On day 1 of CFC, we recorded neuronal activities during footshocks and compared the firing rates (FR) during the 2 sec before and during footshocks (Fig. 5A). Then, we calculated the shock response scores (SRC, see Methods) for each neuron with scores of 1.0 and −1.0 being maximally activated and inhibited, respectively (Fig. 5B, C). To classify footshock-positive responsive neurons (pro-footshock) or footshock-negative responsive neurons (anti-footshock), we generated a null SRC distribution from the mean SRC for each shuffled spike train (see method for details). A neuron whose mean SRC was larger than the top 2.5% of the null SRC

distribution was considered pro-footshock and a neuron with a mean SRC lower than the bottom 2.5% of the null distribution was considered anti-footshock neuron. This analysis revealed that the fraction of pro-footshock neurons was much larger in the BLA$^{Etv1}$ population (42.2%) than in the BLA$^{Lypd1}$ population (28.6%) (Fig. 5C).

On day 2 of CFC, we monitored the contextual freezing response, which was similar between the two populations of mice (Supplementary Fig. 10I). When we correlated the frequency of freezing of individual mice with the percentage of pro-footshock neurons on day 1, we found a positive although not statistically significant trend in the BLA$^{Etv1}$ population ($R^2 = 0.6$), but no such trend in the BLA$^{Lypd1}$ population (Fig. 5D). These results indicate that a sizeable fraction of BLA$^{Etv1}$ neurons was activated by innate fear stimuli and raise the possibility that the fraction of pro-footshock neurons contributes to the conditioned freezing response.

## Activities of BLA$^{Etv1}$ neurons increase during social interactions

Next, we examined whether these three neuron populations were modulated by social interactions, a type of consummatory behavior that was previously shown to be regulated by the BLA but was not associated with a specific neuron population[18,23,31]. We confronted individual mice with a younger conspecific of the same gender confined in a wired container (social affective paradigm), either in a round cage or a two-compartment chamber (Fig. 5E). Social behavior was quantified as the approach behavior towards the other mouse using data from both chambers. The area in which we observed social interactions, was within a 10 cm radius around the center of the wired container and was termed the "pro-social area". The area in which no approach behavior or social interactions occurred was outside a 20 cm radius around the wired container and was termed the "anti-social area". The area in between was termed the neutral area (Fig. 5E, F). Similar to the analysis of neuronal activity during the food approach, we performed permutation tests to identify the neurons whose activity was significantly correlated with the distance to the social interaction partner and found that the social-distance sorting pattern of the significant neurons was consistent with the pattern of all neurons (Fig. 5G–I).

The quantification of all neurons revealed that the smallest fraction of active BLA$^{Lypd1}$ neurons was in the pro-social area (24.6%), both for all and significant neurons (Fig. 5J), and this percentage value was significantly lower than the chance level in comparison with the shuffled data of significant neurons from all three populations (Fig. 5J, right). Interestingly, the largest fraction of active neurons in the BLA$^{Etv1}$ population was in the pro-social area (42.6%) and this percentage value was significantly higher than the chance level (Fig. 5K). In contrast, the percentages of active BLA$^{Rspo2}$ neurons did not change in this social interaction assay (Fig. 5L). The total distance moved during social tasks was similar across Cre lines (Supplementary Fig. 10J). These results indicated that BLA$^{Etv1}$ neurons were activated during social interactions.

The response properties of BLA$^{Etv1}$ neurons raised the question of whether the same neurons were activated during social interactions and footshocks or this population was more heterogeneous than suggested by transcriptomics data. To monitor the same neurons for both social interaction and footshock responses, we performed the social interaction test in the fear conditioning chamber. Calcium traces were recorded in the presence of a wired container with a novel mouse during a "social interaction period". Then, the container was taken out and five consecutive footshocks were delivered (Supplementary Fig. 11A, B). Example traces of neurons active during social interactions and/or footshocks are shown in Supplementary Fig. 11C. Since the fear conditioning chamber was smaller than the chamber previously used for social interactions, we scored the social response score (SoRC) differently (see methods). Animals were considered to be engaged in social interactions when their heads were oriented towards the wired container and their distance to the center of the container was less than 10 cm. To calculate the SoRC, we deconvolved calcium traces to

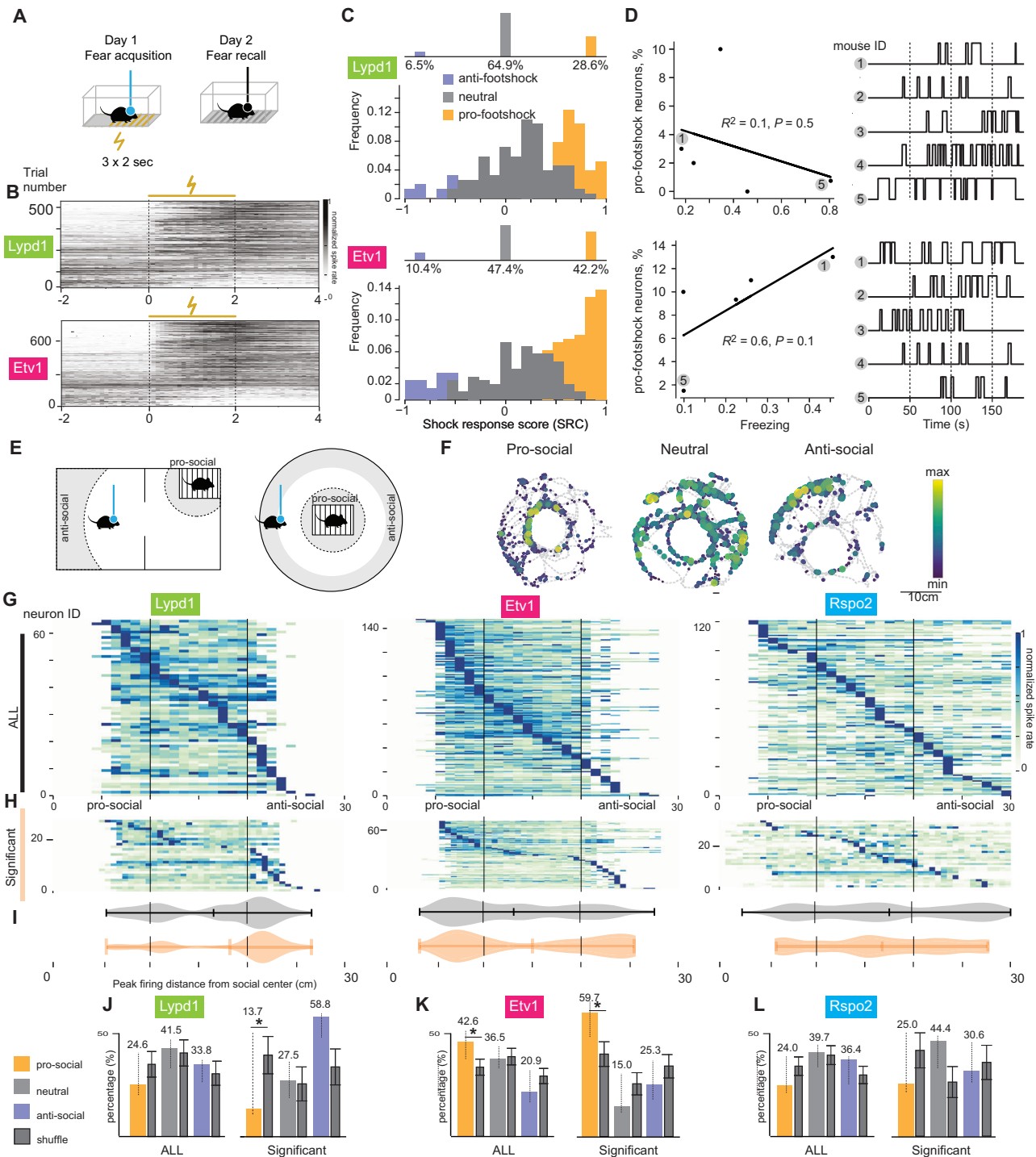

**Fig. 5 | Genetically and spatially defined neurons show different responses to aversive and social cues. A** Scheme of contextual fear conditioning assay (CFC). **B** Peri-stimulus time histogram (PSTH) illustrating inferred spike trains from calcium responses during the footshock session of CFC in Lypd1 (top) and Etv1 (bottom) mice. Trial numbers are neuron numbers multiplied by three footshocks, sorted by footshock response score (SRC). **C** Histogram of SRC in BLA[Lypd1] (top) and BLA[Etv1] (bottom) mice showing negative correlation (anti-footshock, purple), positive correlation (pro-footshock, orange), and neutral (gray). ($N = 179$ and 211 neurons recorded from 7 and 6 mice, BLA[Lypd1], BLA[Etv1] respectively). **D** Scatter plots show the relationship between freezing frequency during fear retrieval and percentages of pro-footshock neurons in fear acquisition for Lypd1 (top) and Etv1 (bottom) mice ($n = 5$ mice each, the corresponding $R^2$ and $p$-values from the two-sided test). The plots on the right show the binarized freezing traces for each mouse.

**E** Schemes of social interaction assays (two chamber or round chamber with conspecific). **F** Response maps of three example neurons (pro-social, anti-social, neutral). **G, H** Averaged firing rate heatmaps to social distance during the social interaction assay in all neurons (**G**) or significant neurons determined with a permutation test (**H**) ($N = 64$, 148, and 121 neurons recorded from 7, 6, and 5 of BLA[Lypd1], BLA[Etv1] and BLA[Rspo2] mice, respectively). **I** Violin plots of peak firing distances (significant neurons: orange; all neurons: gray). **J–L** Percentages of neurons with peak firing rates in pro-social, neutral, and anti-social areas for BLA[Lypd1] (**J**), BLA[Etv1] (**K**) and BLA [Rspo2] (**L**) populations. Significance at the 2.5% level with null distribution (shuffled) is indicated (two-tailed test, mean ± SD). Significant cell numbers are for Lypd1; $n = 4,8,17$ and Etv1; 40,10,17 and Rspo2; 9,16,11 from the pro-food, neutral, and anti-food areas, respectively. Data are presented as mean values SD (**J–L**). *$p < 0.05$, **$p < 0.001$, ****$p < 0.0001$, with all $t$ tests being two-tailed.

firing rates over time and calculated the average firing rates for the social-interaction phases and non-social phases for each neuron. Neurons with a positive SoRC were termed 'social ON' neurons, while those with a negative SoRC were termed 'social OFF' neurons. The shock response score (SRC) was calculated as described in Fig. 5 and the method section. However, we did not classify the footshock-responsive neurons using the permutation test but rather termed neurons with positive SRC 'shock ON' neurons and those with negative SRC 'shock OFF' neurons.

The quantification revealed that the largest fraction of BLA$^{Etv1}$ neurons (44%) were both 'social ON' and 'shock ON' neurons and outnumbered the fractions that were either 'social ON' or 'shock ON' neurons (16 and 22%, respectively; Supplementary Fig. 11D). Linear regression analysis revealed a significant positive correlation in the SRCs and SoRCs in BLA$^{Etv1}$ neurons ($p = 0.01$) although the data showed high variability with low $R^2$ value (Supplementary Fig. 11E). In comparison, the 'social ON' and 'shock ON' fraction of BLA$^{Lypd1}$ neurons was rather small (21%, Supplementary Fig. 11F), consistent with the results on social interactions in Fig. 5. Moreover, linear regression analysis revealed a lack of positive correlation in the SRCs and SoRCs in BLA$^{Lypd1}$ neurons ($p = 0.41$; Supplementary Fig. 11G). These results confirm that the BLA$^{Etv1}$ population contains a large fraction of neurons that are activated during social and defensive behavior.

## BLA$^{Lypd1}$ neurons are positive valence neurons and promote normal food uptake

The activation patterns of these BLA neurons suggested that they participated in valence-specific behaviors. We first asked if optogenetic activation of these populations would be sufficient to promote appetitive behavior. We also employed optogenetic inhibition approaches to investigate, if one or more of these populations would be necessary to mediate appetitive behavior. We expressed channelrhodopsin-2 (ChR2) in all three Cre lines using a Cre-dependent viral vector (AAV5-Ef1α-DIO-hChR2(H134R)-EYFP) bilaterally targeted to the BLA and implanted optical fibers bilaterally over the BLA for somata photostimulation (Fig. 6A, B). Control mice received a similar AAV vector lacking ChR2 (AAV5-Ef1-DIO- EYFP). The feeding assay was the same as the one used for calcium imaging and included an approach to the food, subsequent lifting/handling, and ingestion of the pellet. Food consumptions during light-On and light-Off phases were measured on separate days using the same cohorts of mice. After 20 h of fasting, photoactivated Lypd1-Cre::ChR2 mice consumed significantly more food than EYFP control mice and in comparison to Light-off days (Fig. 6C). This was in contrast to Etv1-CreER::ChR2 and Rspo2-Cre::ChR2 mice, which consumed significantly less food during the Light-On compared to the Light-Off phase (Fig. 6C). The observed effects were independent of general locomotor behaviors (Supplementary Fig. 12A). To acutely photoinhibit neurons, we expressed Cre-dependent Halorhodopsin (eNpHR3.0-mCherry) in a similar fashion as ChR2 and assessed food consumption. We found that photoinhibited, hungry Lypd1-Cre::eNpHR mice ate significantly less food than in the absence of photoinhibition (Fig. 6D), while the same manipulation had no effect on Etv1-CreER::eNpHR and Rspo2-Cre::eNpHR mice. In summary, the activity of BLA$^{Lypd1}$ neurons is both sufficient and necessary to promote feeding. Activation of BLA$^{Etv1}$ or BLA$^{Rspo2}$ neurons can suppress feeding. However, these neurons may not be required for food uptake in the free-feeding assay.

We also assessed the valence of optogenetic activation and inhibition of the three types of BLA neurons in the neutral environment of a conditional place preference assay (CPP) (Fig. 6E, Supplementary Fig. 13G, see Methods). After conditioning, Lypd1-Cre::ChR2 mice exhibited a significant preference for the photostimulation-paired chamber, whereas Etv1-CreER::ChR2 and Rspo2-Cre::ChR2 mice showed significant avoidance behavior for the photostimulation-paired chamber (Fig. 6E). Conversely, photoinhibition of BLA$^{Lypd1}$

neurons caused a significant avoidance of the photostimulation-paired chamber, whereas photoinhibition of BLA$^{Etv1}$ neurons resulted in significant preference behavior for the photostimulation-paired chamber (Supplementary Fig. 13G). No changes in anxiety-like behavior were observed in Open-Field behavior (Supplementary Fig. 12B). These results indicate that mice can learn to associate an open area with positive valence for photoactivation of BLA$^{Lypd1}$ neurons and conversely, with negative valence for photoactivation of BLA$^{Etv1}$ or BLA$^{Rspo2}$ neurons.

## BLA$^{Etv1}$ neurons are necessary for fear memory formation

Given that BLA$^{Etv1}$ neurons were strongly activated by footshocks, we next asked if optogenetic manipulation of these neurons would affect the freezing response in a CFC experiment. On day 1 of CFC, footshocks were paired with either photoactivation or photoinhibition of the somata of BLA$^{Etv1}$ or BLA$^{Lypd1}$ neurons (Fig. 7A, B). On day 2 (Fear recall), the fraction of time the animals spent freezing was monitored. Photoactivation of Lypd1-Cre::ChR2 mice resulted in significantly less freezing than photoactivation of EYFP control mice, while similar levels of freezing were observed in Etv1-CreER::ChR2 mice compared to their respective EYFP control mice (Fig. 7A). Conversely, photoinhibition of Etv1-CreER::eNpHR mice resulted in significantly less freezing on fear recall day compared to their respective mCherry control mice, while photoinhibition of Lypd1-Cre::eNpHR mice resulted in an increase of freezing compared to Lypd1-Cre::mCherry control mice (Fig. 7B). The reduction in freezing of photoinhibited Etv1-CreER::eNpHR mice could already be observed during fear acquisition (day 1) (Supplementary Fig. 12D). These results showed that BLA$^{Etv1}$ neurons are necessary for fear memory formation. They further indicate that BLA$^{Lypd1}$ neurons are sufficient to suppress freezing behavior.

## BLA$^{Etv1}$ neurons are necessary for social interaction

Given that a large fraction of BLA$^{Etv1}$ neurons were activated during social behavior, we next asked, if optogenetic manipulation of these and other neurons would alter social behavior. Social behavior assays were performed for calcium imaging experiments. Interestingly, photoactivation of Etv1-CreER::ChR2 mice resulted in mice spending more time in the social zone compared to the light-off phase, an effect that was not observed in control mice expressing YFP (Fig. 7C). Instead, interactions of photoactivated Etv1-CreER::ChR2 mice with the empty cage were unaffected (Supplementary Fig. 12E). The converse effect was observed in photoinhibited Etv1-CreER::eNpHR mice which spent significantly less time in the social zone compared to the light-off phase (Fig. 7D). Neither optogenetic manipulation of BLA$^{Lypd1}$ nor BLA$^{Rspo2}$ neurons altered their social behavior, which was in line with the observed neutral responses in the calcium imaging experiments. Additionally, there was no indication of sex differences in optogenetic experiments regarding social interactions, nor in any of the other behavioral experiments (Supplementary Fig. 13 A–C). When comparing the results across genotypes, we found that social interactions and freezing behavior were significantly promoted by BLA$^{Etv1}$ neuron activation in comparison to BLA$^{Lypd1}$ neuron activation, and similar effects were observed by BLA$^{Lypd1}$ neuron inhibition compared to BLA$^{Etv1}$ neuron inhibition. The opposite effects across genotypes were seen for the promotion of food consumption (Supplementary Fig. 13D–F). Together, these results showed that BLA$^{Etv1}$ neurons encoded sociability and were sufficient and necessary to drive social interaction.

## Discussion

In this report, we have described a full single-cell transcriptomic analysis of glutamatergic neurons in the BLA of adult mice. In combination with smFISH, we characterized a total of 11 cell clusters and demonstrated that they were distributed in distinct spatial BLA subregions. Several clusters showed a clear preference between LA and BA, and other clusters were enriched in either anterior or posterior regions of

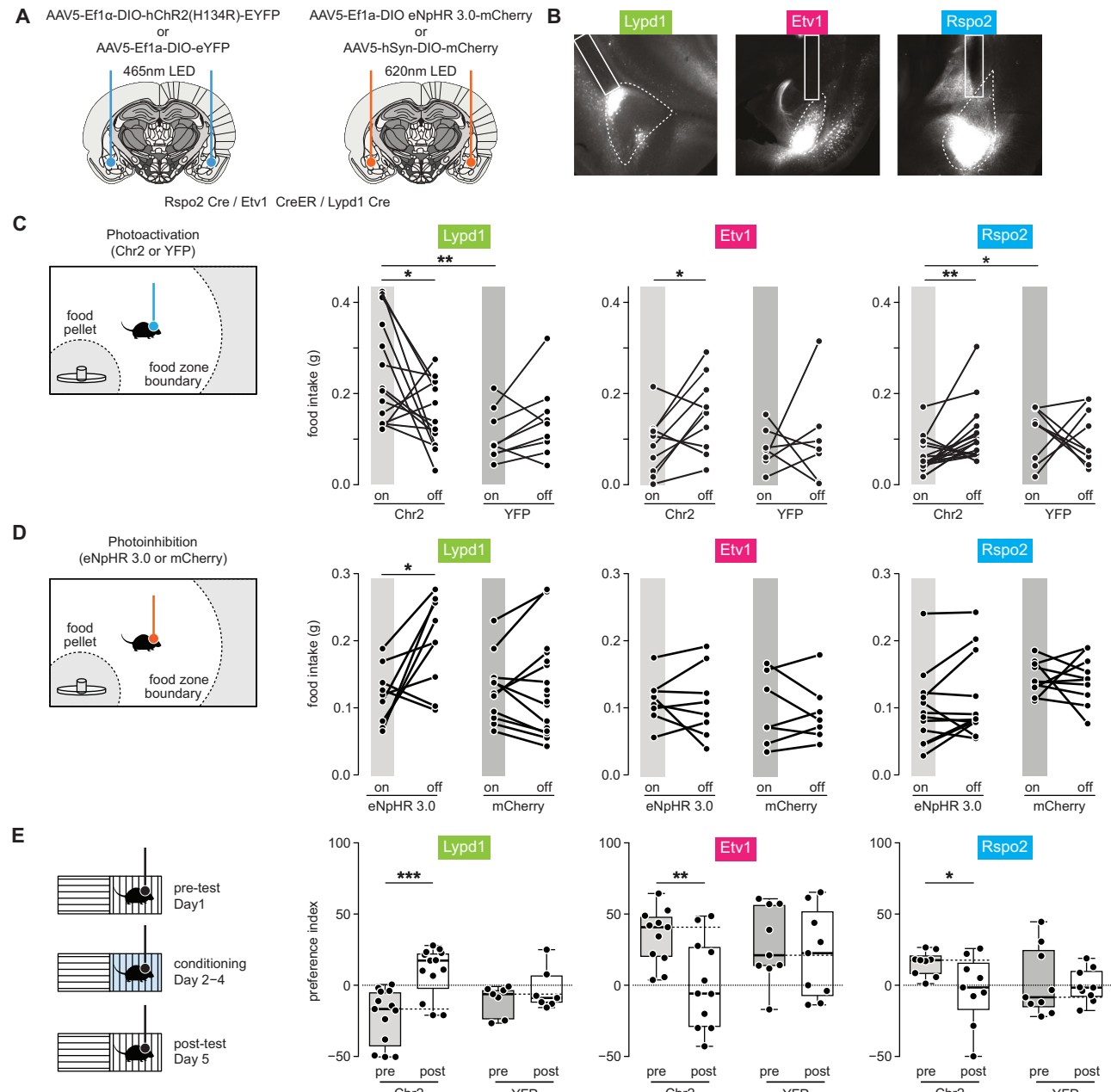

**Fig. 6 | BLA^Lypd1 neurons promote normal feeding behavior. A** Schemes of AAV injections and optic-fiber placements above ChR2- and eNpHR-expressing BLAs; modified from Allen Mouse Brain Atlas, mouse.brain-map.org. **B** Representative images of ChR2-eYFP expression in Rspo2, Etv1 and Lypd1-Cre mice with optic fiber locations. **C** Left: Scheme of optogenetic activation during the free-feeding assay. Right: Food intake during optogenetic activation of three BLA populations compared to light-off epochs and photostimulated controls. Lypd1: $n = 14$ (ChR2) and 9 mice (YFP) per group with a two-tailed paired $t$ test, $t_{(13)} = 2.457$, $p = 0.0288$ within ChR2 (on versus off). For ChR2-On versus YFP-On: two-tailed unpaired $t$ test, $t_{(21)} = 3.4$, $p = 0.0027$. Etv1: $n = 11$ (ChR2) and 7 mice (YFP) per group with a two-tailed paired $t$ test, $t_{(9)} = 2.492$, $p = 0.0343$ within ChR2 (on versus off). Rspo2: $n = 18$ (ChR2) and 9 mice (YFP) per group with Wilcoxon matched pairs signed rank test, $p = 0.0023$ within ChR2 (on versus off) group. For ChR2-On versus YFP-On: Kolmogorov–Smirnov test, $p = 0.0226$. **D** Left: Scheme of optogenetic inhibition. Right: Food intake during optogenetic inhibition of three BLA populations compared to light-off epochs and photostimulated controls. Lypd1: $n = 9$ (eNpHR 3.0) and 11 mice (mCherry) per group with two-tailed paired $t$ test, $t_{(8)} = 2.771$, $p = 0.0243$ within eNpHR 3.0 (on versus off) group. Etv1: $n = 8$ (eNpHR 3.0) and 7 mice (mCherry) per group. Rspo2: $n = 13$ (eNpHR 3.0) and 10 mice (mCherry) per group. **E** Left: Scheme of conditioned-place preference experiment. Right: Preference index (cumulative time % in paired chamber−cumulative time % in unpaired chamber) of cohorts of mice before (pre) and after (post) conditioning. Lypd1: $n = 13$ (ChR2) and 7 mice (YFP) per group; paired $t$ test, $t_{(12)} = 4.528$, $p = 0.0007$ within ChR2 group (pretest versus posttest), Etv1: $n = 11$ (ChR2) and 9 mice (YFP) per group; paired $t$ test, $t_{(10)} = 3.273$, $p = 0.0084$ within ChR2 group (pretest versus posttest). Rspo2: $n = 8$ mice (ChR2 and YFP) per group; two-tailed paired $t$ test, $t_{(7)} = 2.695$, $p = 0.0308$, within ChR2 group (pretest versus posttest). All box-and-whisker plots show whiskers down to the minimum and up to the maximum value and boxes with center (median) and bounds of the box (75th and 25th percentile). *$p < 0.05$, **$p < 0.01$, ***$p < 0.001$. All $t$ tests used two-tailed.

the BLA. We selected three genetic markers for functional analysis. *Lypd1*, the marker with highest expression in LA and additional regions in BA, targeting three transcriptomic clusters (cl2,8,9); *Etv1*, the marker with scarce expression in LA, complementary pattern with *Lypd1* in BA

subregions, targeting three clusters (cl1,4,9); *Rspo2*, scarce expression in LA, restricted pattern in anterior BA, partially overlapping with *Etv1*, targeting two clusters (cl1,10). We found that BLA^Lypd1 neurons are positive-valence neurons: they are activated during fasting and food

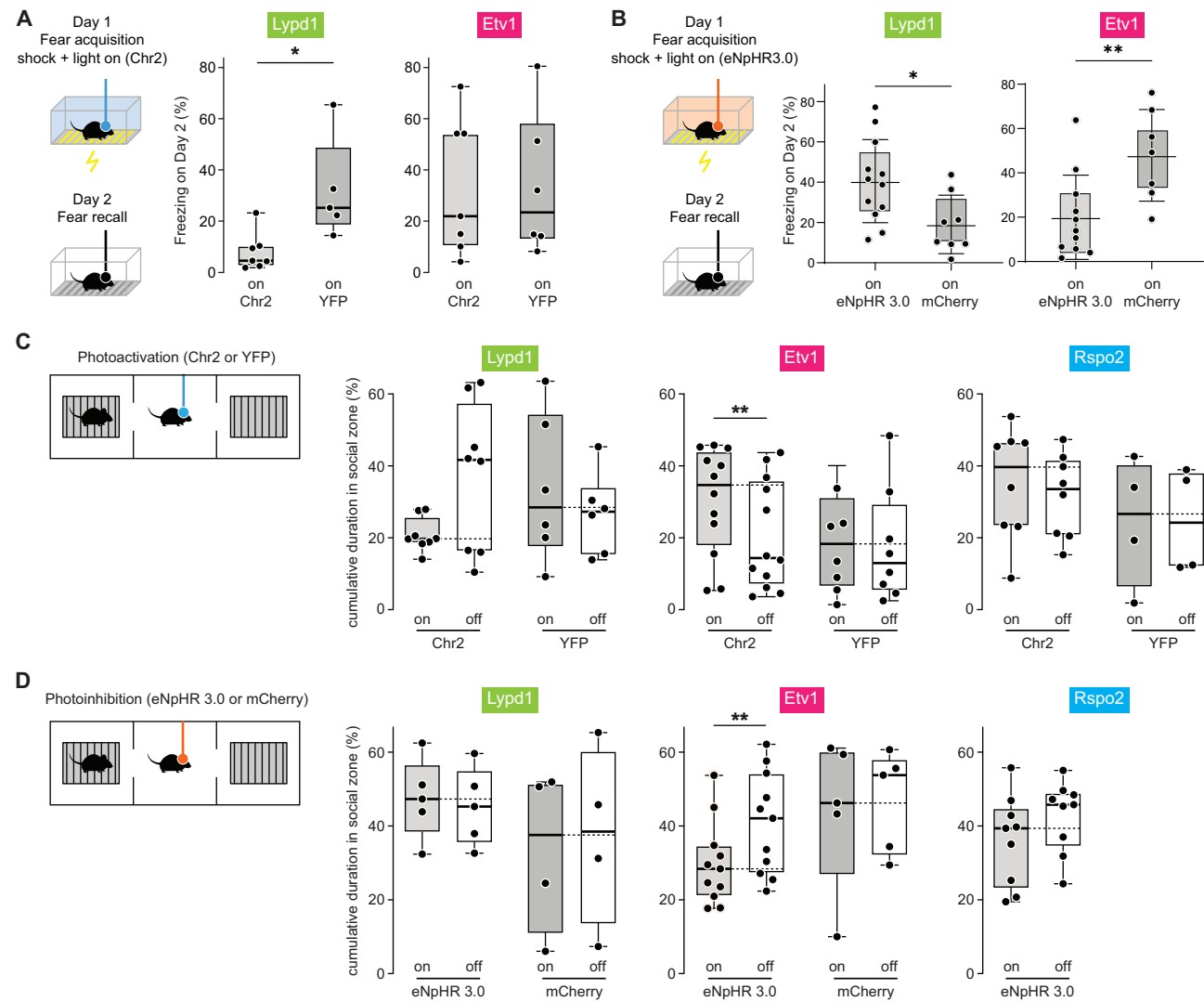

**Fig. 7 | BLA^Etv1 neurons are necessary for fear memory formation and social interactions. A** Left: Scheme of contextual fear conditioning with photostimulation; Day 1: 3 footshocks (0.75 mA) paired with light on. Freezing measured on Day 2 (fear recall). Right: Freezing behavior (%) on Day 2 with photostimulation of two BLA populations compared to controls. Lypd1: $n = 7$ (ChR2) and 5 mice (YFP) per group; Kolmogorov–Smirnov test, $p = 0.0152$, Etv1: $n = 6$ (ChR2) and 7 (YFP) mice per group; Kolmogorov–Smirnov test, $P = 0.9254$. **B** Left: Scheme of contextual fear conditioning with photoinhibition; Right: Freezing behavior (%) on day 2 combined with photoinhibition of two BLA populations compared to controls. Lypd1: $n = 12$ (eNpHR 3.0) and 8 (mcherry) mice per group; unpaired $t$ test, $p = 0.0204$, *$p < 0.05$. Etv1 groups: $n = 11$ (eNpHR 3.0) and 7 (mcherry) mice per group; unpaired $t$ test, $p = 0.0096$. **C** Schemes of social interaction assay with photoactivation; Right: Cumulative duration in the social zone (%) combined with photoactivation of three BLA populations compared to light-off epochs and controls. Lypd1; $n = 8$ (ChR2) and 6 mice (YFP) per group; two-tailed paired $t$ test, $t_{(7)} = 2.307$, $p = 0.0544$ within

ChR2 group (on versus off); Etv1; $n = 12$ (ChR2) and 8 mice (YFP) per group; paired $t$ test, $t_{(11)} = 3.785$, $p = 0.0030$, within ChR2 group (on versus off); Rspo2; $n = 6$ (ChR2) and 4 mice (YFP) per group; paired $t$ test, $t_{(7)} = 0.6806$, $p = 0.5180$, within ChR2 group (on versus off). **D** Schemes of social interaction assay with photoinhibition; Right: Cumulative duration in the social zone (%) combined with photoinhibition of three BLA populations compared to light-off epochs and controls. Lypd1; $n = 5$ (eNpHR3.0) and 4 mice (mcherry) per group; Wilcoxon matched-pairs signed rank test, $p = >0.9999$, within eNpHR 3.0 group (on versus off); Etv1; $n = 11$ (eNpHR3.0) and 5 mice (mcherry) per group; paired $t$ test, $t(10) = 3.19$, $p = 0.0097$, within eNpHR 3.0 group (on versus off); Rspo2; $n = 9$ (eNpHR3.0) mice, Wilcoxon matched-pairs signed rank test, $p = 0.6250$, within eNpHR 3.0 group (on versus off). All box-and-whisker plots show whiskers down to the minimum and up to the maximum value and boxes with center (median) and bounds of the box (75th and 25th percentile with all $t$ tests being two-tailed. *$p < 0.05$, **$p < 0.01$, ***$p < 0.001$.

approach behavior, and mediate normal food uptake. BLA^Etv1 neurons are mixed-selectivity neurons: they are activated by innate fear stimuli and during pro-social approach behavior. They promote the formation of fear memory and promote social interactions. BLA^Rspo2 neurons are negative-valence neurons: BLA^Rspo2 match the idea of a typical aversive population by not responding to either food or social cues and promoting aversive conditioning. Together these findings describe the rich diversity of glutamatergic cell types and their spatial distribution in the BLA, support the concept that genetically defined subpopulations respond either to valence-specific or mixed cues, and expand

their behavioral output to the promotion of normal feeding and social behavior.

Our single nuclei transcriptomics analysis provides an extensive account of glutamatergic cell clusters. The results matched pretty well with previously published work[43] based on Seurat::IntegrateData. For example, our cluster 8: (Rorb/Lypd1) which was enriched in anterior LA, showed similarity with O'Leary et al.'s LA1 subpopulation (Supplementary Fig. 14A, B). Cluster 2, (Sema5a/Otof/Lypd1/Cdh13) which localized to posterior LA, matched mainly with O'Leary et al.'s LA2 subpopulation (Supplementary Fig. 14B). Compared to LA, the BA

clusters matched less well. Cluster 1 (Rspo2/Etv1/Adamts2/Bdnf), which localized to the whole anterior-posterior extent of the BA, was included in all O'Leary et al.'s BA1-BA4[43]. Importantly, our most anterior and most posterior located clusters did not match, suggesting that the very anterior and very posterior extents of the BLA were not included in the O'Leary dataset[43] (Supplementary Fig. 14B). We also compared our data with a scRNAseq study comprising the whole amygdala[47]. When we integrated all of the excitatory BLA clusters from this study with our data, several of the main clusters matched well (Supplementary Fig. 15A, B). For example, our BA cluster 1 (Rspo2/Etv1/Adamts2/Bdnf) showed a near-perfect match with Hochgerner's VGLUT1-2 cluster from anterior BA (Supplementary Fig. 15C, D). Our clusters from the LA (cluster 2, Sema5a/Otof/Lypd1/Cdh13 and cluster 8, Rorb/Lypd1) showed similarity to Hochgerner's VGLUT1-28 and VGLUT1-29 clusters from LA (Supplementary Fig. 15C, E). A comparison of genes matching in both datasets revealed that the pattern of gene expression was distinct between the BA and LA clusters supporting the idea that these subregions contain genetically defined subpopulations (Supplementary Fig. 15D, E). Some of our most anterior and most posterior located clusters (our clusters 5 and 7) remained unassigned in the Hochgerner subset, possibly because they were assigned to other amygdala clusters. Another recent BLA snRNAseq study comparing BLA cell types with published work confirmed a good match with our data[53].

Our characterization of BLA[Rspo2] neurons corresponds well with previous work[33] in terms of their enrichment in the anterior BA and their function in negative-valence behavior. The same study characterized BLA[Ppp1r1b] as positive-valence neurons being spatially segregated from BLA[Rspo2] neurons in the posterior BA[33]. More recent studies challenged this view by showing that the majority of Rspo2+ neurons in the BLAp were Ppp1r1b+[34] and that Ppp1r1b is a marker for GABAergic clusters[47]. Our snRNAseq dataset did not include *Ppp1r1b*, perhaps reflecting limitations of snRNAseq in detecting certain low abundant mRNAs. *Fezf2*, another marker for valence-specific BLA neurons[34] was expressed rather widely in our snRNAseq dataset, including BLA[Rspo2] neurons, and may represent a rather large fraction of BA neurons with functional heterogeneity.

The present study characterized distinct cell clusters in LA and BA which confirms the O'Leary et al. dataset[43], and is consistent with the LA having a distinct role from the BA in emotional learning[54]. Previous work indicated the presence of continuous spatial gene-expression gradients[33,43,55], rather than distinct cell clusters that are positioned in specific spatial locations. We used two different methods to assign spatial locations to cell clusters. First, we assigned 10 specific marker genes to represent all cell clusters and delineated their expression within eight previously published subregions using smFISH. Second, we used the smFISH data to map every cell belonging to a specific transcriptional cluster to BLA space in an unsupervised manner. Using both methods we find evidence for both scenarios, clusters with graded variability (e.g. cluster 1: (Rspo2/Etv1/Adamts2/Bdnf) in BA) and clusters showing more distinct spatial locations (e.g. cluster 3: (Otof), no expression in aBA, high in pBA, or cluster 7: (Rorb) high in aLA, no expression in pLA.

Our functional analysis revealed the presence of a previously uncharacterized BLA[Lypd1] neuron population as positive-valence neurons. BLA[Lypd1] neurons are found in three cell clusters in LA and acBA, complementary to BLA[Etv1] neurons which are enriched in amBA and alBA. In slice electrophysiological recordings, BLA[Lypd1] neurons showed increased firing rates in fasted animals, similar to our previous recordings of appetitive CeA neurons[56]. In-vivo calcium recordings revealed that a large fraction of BLA[Lypd1] neurons are active during food approach behavior, whereas smaller fractions of BLA[Lypd1] neurons respond to fear and social stimuli. Optogenetic manipulations demonstrated that BLA[Lypd1] neurons are both sufficient and required for normal food uptake. Although we did not provide direct data for

BLA[Lypd1] neurons being activated during eating, the fact that the activity of BLA[Lypd1] neurons increases when the animal is in the food zone suggests exactly that. Activation of BLA[Lypd1] neurons during fasting was observed in tissue slices, where Lypd1 neurons showed higher excitability compared to slices from fed mice. Our interpretation of these data is that palatable food engages BLA[Lypd1] neurons. The hungrier the animals are, the more palatable the food becomes. During fasting, BLA[Lypd1] neurons are activated and contribute to the process of palatability-guided feeding. BLA[Lypd1] neurons therefore have similar roles as recently described central amygdala appetitive neurons[56,57]. These results are surprising in light of previous work indicating that the BLA does not promote free-feeding behavior[26,27], and that it has a negative effect on food consumption in a risky environment[28]. The main difference to previous work is that we have manipulated a distinct subpopulation, enriched in LA and certain parts of BA, whereas previously either most of the BA or certain BLA projections had been manipulated. Hence, BLA[Lypd1] neurons may have largely been left untouched in previous manipulation experiments. The role of BLA PN in promoting normal feeding is supported by studies that have shown that BLA neurons are responsive to a variety of foods[29,30]. The positive valence function of BLA[Lypd1] neurons is not restricted to feeding, since mice can be conditioned to associate a positive valence with the photoactivation of BLA[Lypd1] neurons in a CPP assay and since the photoactivation of BLA[Lypd1] neurons suppresses the formation of fear memory. The function of these neurons is, however, not to enhance any ongoing motivated behavior, since BLA[Lypd1] neurons do not modulate social interactions. The mechanism by which BLA[Lypd1] neurons promote food intake remains to be explored. It is possible that BLA[Lypd1] neurons directly synapse onto orexigenic CeA neurons including the recently described Ghrelin-responsive CeA[Htr2a] neurons[56]. It also remains to be tested whether the activity of BLA[Lypd1] neurons is modulated by other internal states besides hunger and by environmental factors such as risk stimulus[28]. It may be interesting for human studies to investigate BLA's role in the prevalence of maladaptive eating behaviors in humans.

While BLA[Lypd1] neurons are mainly positive-valence neurons, BLA[Etv1] neurons show flexible mixed selectivity and are activated by aversive cues (foot shock) and during social interactions, but not during fasting or by the presence of food. This observation supports previous studies showing that BLA neurons can respond to multiple stimuli, including social and non-social cues[25,39,58–61]. In line with the calcium imaging data, our optogenetic manipulation experiments revealed that BLA[Etv1] neurons are mainly negative valence neurons promoting defensive behavior (contextual fear memory formation and conditional place aversion) and reducing food intake, in line with previous reports on BLA functions[35,36,62–64]. However, BLA[Etv1] neurons also promote social behavior which we interpret as being rewarding, a surprising observation, considering that most of the previous evidence suggested that the BLA mediates aversive aspects of social interaction[19,23]. A recent report paints a more detailed picture by showing that medial prefrontal cortex to BLA subcircuits regulate social preference in a bi-directional manner. While prelimbic cortex (PL) -BLA projectors suppress, infralimbic cortex (IL) -BLA projectors were required for social interactions[21]. These results raise the interesting possibility that the target neurons of IL-BLA projectors may be BLA[Etv1] neurons. A recent study identified a subpopulation of BLA PN expressing secretin (SCT) as promoters of social behavior[18] indicating the presence of distinct subsets of BLA neurons facilitating social interactions. Whether BLA[SCT] neurons overlap with BLA[Etv1] neurons and whether they also promote fear memory formation remains to be investigated. Our calcium imaging data indicates that BLA[Etv1] neurons consist of more than one population, one responding to shock, one to social cues, and one—the largest one—to both. Part of the mixed selectivity of these neurons may arise from the fact that optogenetic manipulation will affect all populations together. Further work is

therefore needed to functionally characterize the BLA[Etv1] subpopulations and their associated microcircuits which are likely very complex. Having access to genetically defined subpopulations of BLA neurons that are integrated into these circuits will greatly accelerate the process of social circuit dissection. Similar to the regulation of food consumption, social behavior and the underlying amygdala circuits are evolutionarily conserved. Their examination of different animal models and humans will ultimately benefit our understanding of the circuit basis of psychiatric disorders.

## Methods

### Animals

Experiments were performed using adult mice (>8 weeks). The wild-type animals were from the C57BL/6NRj strain (Janvier Labs). The Rspo2-Cre transgenic line (C57BL/6J-Tg(Rspo2-cre)Blto (RBRC10754)) from RIKEN BioResource Research Center) and Etv1-CreER transgenic line (Etv1tm1.1(cre/ERT2)) from Jackson Laboratory) and Lypd1-Cre (Tg(Lypd1-cre)SE5Gsat/Mmucd) mice were imported from the Mutant Mouse Regional Resource Center). Td-Tomato Rosa26R mouse lines were as described previously[65], using the line Ai9lsl-tdTomato [B6.Cg-Gt(ROSA)26SorTM9.CAG-tdTomato/Hze/J][66]. Transgenic mice were backcrossed with a C57BL/6N background. Animals used for optogenetic manipulations and calcium imaging were handled and singly housed on a 12 h inverted light cycle for at least 5 days before the experiments. Mice were given ad libitum food access except during food deprivation for feeding experiments. All behavior assays were conducted at a consistent time during the dark period (2 p.m.–7 p.m.). Both male and female mice were used and all the experiments were performed following regulations from the government of Upper Bavaria.

### Viral constructs

The following adeno-associated viruses (AAVs) were purchased from the University of North Carolina Vector Core (https://www.med.unc.edu/genetherapy/vectorcore): AAV5-ef1a-DIO-eNpHR3.0-mCherry, AAV5-hSyn-DIO-mCherry, AAV5-Ef1α-DIO-hChR2(H134R)-EYFP, AAV5-Ef1a-DIO-eYFP. The AAV5.Syn.Flex.GCaMP6f.WPRE.SV40 virus was obtained from Addgene.

### SnRNA-seq

Single-nucleus RNA sequencing was focused on the basolateral amygdala (BLA). For visually guided dissection of the BLA, we practiced by using the fluorescent expression in basal amygdala (BA) of Rspo2-cre; tdTomato mice[33] and dense fiber tracks surrounding BLA boundaries to facilitate complete microdissection of BLA. Each single nucleus sequencing dataset includes BLA tissues from 4 male brains (both hemispheres). To reduce potential batch effects, brains were always from the same litter, collected and processed in parallel at the same time. Mice were deeply anesthetized by i.p. injections of 200 mg/kg Ketamine and 40 mg/kg Xylazine, and perfused with 10 mL ice-cold Sucrose-HEPES Cutting Buffer containing (in mM) 110 NaCl, 2.5 KCl, 10 HEPES, 7.5 MgCl2, and 25 glucose, 75 sucrose (~350 mOsm/kg), pH = 7.4[67]. All the solutions/reagents were kept on ice in the following procedures unless otherwise specified. The brain was extracted and cut (300 µm) on a vibratome (Leica VT1000S, Germany) in cutting buffer, and the slices were transferred into a Dissociation Buffer containing (in mM): 82 Na2SO4, 30 K2SO4, 10 HEPES, 10 glucose and 5 MgCl2, pH = 7.4[67]. BLA was microdissected under a microscope (Olympus SZX10) covering the anterior (Bregma −0.59) and posterior (Bregma −3.0) extent of the adult BLA.

### Single nucleus isolation and library preparation

The protocol for single nucleus isolation was optimized from previous studies[68,69] and demonstrated nucleus isolation protocol (CG000393, 10x Genomics). In brief, collected tissue chunks from the four brains were transferred in 600 µl homogenization buffer containing 320 mM sucrose, 5 mM CaCl2, 3 mM Mg (CH3COO)2, 10 mM Tris HCl pH 7.8, 0.1 mM EDTA pH 8.0, 0.1% NP-40 (70% in H2O, Sigma NP40S), 1 mM β-mercaptoethanol, and 0.4 U/µl SUPERase RNase inhibitor (Invitrogen AM2694). The homogenization was performed in a 1 mL Wheaton Dounce tissue grinder with 20 strokes of loose and then 20 strokes of a tight pestle. The homogenized tissue was filtered through a 20-µm cell strainer (Miltenyi Biotec) and mixed with an equal volume of working solution containing 50% OptiPrep density gradient medium (Sigma-Aldrich), 5 mM CaCl2, 3 mM Mg (CH3COO)2, 10 mM Tris HCl pH 7.8, 0.1 mM EDTA pH 8.0, and 1 mM β-mercaptoethanol. The resulting solution was transferred into a 2 mL centrifuge tube. A 29% OptiPrep density gradient solution including 134 mM sucrose, 5 mM CaCl2, 3 mM Mg (CH3COO)2, 10 mM Tris HCl pH 7.8, 0.1 mM EDTA pH 8.0, 1 mM β-mercaptoethanol, 0.04% NP-40, and 0.17 U/µl SUPERase inhibitor was slowly placed underneath the homogenized solution through a syringe with a 20 G needle. In the same way, a 35% Density solution containing 96 mM sucrose, 5 mM CaCl2, 3 mM Mg (CH3COO)2, 10 mM Tris HCl pH 7.8, 0.1 mM EDTA pH 8.0, 1 mM β-mercaptoethanol, 0.03% NP-40, and 0.12 U/µl SUPERase inhibitor was slowly laid below the 30% density. The nuclei were separated by ultracentrifugation using an SH 3000 rotor (20 min, 3000g, 4 °C). A total of 300 µl of nuclei was collected from the 29%/35% interphase and washed once with 2 mL resuspension solution containing 0.3% BSA and 0.2 U/µl SUPERase in PBS. The nuclei were centrifuged at 300g for 5 min and resuspended in ~30 µl resuspension solution. Nuclei were stained with DAPI and counted. After manually determining the cell concentration using a hemocytometer, suspensions were further diluted to desired concentrations (300–700 nuclei/µl) if necessary. The appropriate final suspension contained 5000 nuclei and was loaded into the chip. Nanoliterscale Gel Beads-in-emulsion (GEMs) generation, barcoding, cDNA amplification, and library preparation were done using the Chromium Next GEM Single Cell 3′ Reagent Kits v3.1 according to the manufacturer's protocol.

### snRNA-Seq analysis

**Sequence alignment and preprocessing.** Prepared libraries were sequenced on Illumina NextSeq 500 (Mid and High Output Kit v2.5, Paired-end sequencing, 28bp-130bp). We used cellranger (version 7.0.1) to extract fastq files, align the reads to the mouse genome (10x genomics reference build MM10 2020A), and obtain per-gene read counts. Subsequent data processing was performed in R using Seurat (version 4.1.3) with default parameters if not indicated otherwise. After merging the data, we normalized the data (normalization.method = 'LogNormalize', scale.factor=10000), detected variable features (selection.method = 'vst', nfeatures=2000), and scaled the data (vars.to.regress=c('nCount_RNA'). We then applied quality control filters on cells with the following criteria: a) more than 200 genes detected, b) less than 20% mitochondrial gene reads, c) more than 5% ribosomal protein gene reads, d) less than 0.2% hemoglobin gene reads, e) singlets as determined by doubletFinder (version 2.0.3, pK = 0.09, PCs = 1:10). Only genes detected in at least 4 cells were kept. The resulting dataset consisted of 7,953 cells and 21,557 genes. Initial cell clustering was performed with resolution 0.4 after applying harmony batch correction (version 0.1.1) and subsequent UMAP embedding on the harmony reduction.

**Global annotation.** For global annotation, non-neuronal clusters were identified by expression of non-neuronal markers (e.g., *Plp, Mbp, Pdgfra, Olig, Lhfpl3, Igfbp7, Bsg, Tmem119, Cst3, P2ry12, Hexb, C1qb, C1qa, Aldh1l1, Gfap, Slc1a2, Cfap299*) and absence of neuronal markers (*Snap25, Slc17a7, Slc17a6, Neurod6 Syp, Tubb3, Map1b, Elavl2, Gad1, Gad2*, etc.). Neuronal clusters were confirmed by the expression of neuronal markers above and neurotransmitter and neuromodulator releasing neurons were annotated by well-known markers

(glutamatergic neurons: *Slc17a7, Slc17a6, Camk2a, Gria2*, GABAergic neurons: *Adora2a, Gad1, Gad2, Gabbr1, Gabbr1, Gad6S, Gad67*).

For annotation of GABAergic neurons in BLA, firstly only GABAergic neurons based on above global annotation were subtracted and re-clustered. Next, conventional markers from a previous study[70] were used (*Reln, Ndnf, Sst, Pvalb, Vip, Cck, Calb1, Crh, Npy, Foxp2, Htr2a*) Also, GABAergic neuron markers for central amygdala (CEA)[71] were used as reference (*Prkcd, Ppp1r1b, Tac2, Wfs1, Dlk1, Penk, Drd2, Drd1, Calcrl, Pdyn, Nts, Tac1*). We sorted out BLA local inhibitory interneuron from neighborhood regions (e.g., projecting inhibitory neurons in CeA, based on *Pkcd, Drd1* and *Drd2*)[70,71] or intercalated cells (ITCs, based on *Foxp2* expression) or the amygdalostriatal area (based on *Rarb* expression)[72,73].

For clarity, we unified the naming of cell populations in the diverse conditions as follows: clusters from unsupervised clustering were named "Clusters", cell populations containing multiple clusters were named "Category" or named differently.

**Marker-gene selection for glutamatergic neurons for spatial validation (smFISH).** To annotate subtypes of glutamatergic neurons in BLA we retained only glutamatergic neurons and subjected them to re-clustering. Initially, marker genes were identified using presto::top_markers (*n* = 5, auc_min = 0.5, pct_in_min = 20, pct_out_max = 20). We then handpicked the most specific gene for each cluster. For clusters where no marker could be pinpointed, we turned to in situ hybridization (ISH) data from Allen brain atlas: mouse brain. Preference was given to genes that exhibited higher expression in the BLA than other regions and showed localized expression within BLA subregions. Based on these criteria, we selected 10 marker genes shown in Fig. 1F. It is worth noting that other combinations of genes might also adequately represent these molecularly defined cell types.

**Construction of phylogenetic tree of glutamatergic neurons.** Cell type tree was calculated by Seurat::BuildClusterTree on the aggregated expression of all genes using hierarchical clustering of the distance matrix by using Euclidean distance.

**Comparison with published BLA data.** We integrated our mouse basolateral amygdala (BLA) single-cell RNA sequencing (scRNA-seq) data with two published datasets. The first, referred to as the 'O'Leary dataset', includes scRNA-seq data from the mouse BLA (available at GEO:GSE148866)[43] The second, known as the 'Hochgerner dataset', comprises whole amygdala data (available at https://doi.org/10.6084/m9.figshare.20412573)[47]. Using these resources, we constructed a Seurat object for each dataset. The processes of normalization, scaling, and UMAP embedding were performed as previously described in our methods section. Integration of these datasets with our own was accomplished using the 'Seurat::IntegrateData' function in R. This involved the selection of integration anchors, followed by the execution of cell-to-cell mapping using the scmap tool (version 1.18.0), as detailed in the protocol available at https://biocellgen-public.svi.edu.au/mig_2019_scrnaseq-workshop/comparing-and-combining-scrna-seq-datasets.html. Integrated data set was split into two based on origin. The top 5 cluster sharing genes were determined separately and unified for heatmap display.

**HCR sequential multi-fluorescent in situ hybridization.** C57BL/6J mice (*n* = 6, 3 male, 3 female, > 8weeks) were anesthetized IP with a mix of ketamine/xylazine (100 mg/kg and 16 mg/kg, respectively) (Medistar and Serumwerk) and transcardially perfused with ice-cold phosphate-buffered saline (PBS), followed by 4% paraformaldehyde (PFA) (1004005, Merck) (w/v) in PBS. The brain was dissected and immediately placed in a 4% PFA buffer for 2 h at room temperature. The brain was then immersed in 30% RNase-free Sucrose (Amresco, 0335) in 1X

PBS for 48 h at 4 degree until the brain sank to the bottom of the tube. The brain was then embedded in OCT and cryo-sectioned (15 mm thick) by harvesting coronal sections through the AP extent of BLA and stored at −80 °C. At least three coronal sections were selected for analysis within each of the anterior (−0.79 to −1.07 from bregma), anterior-middle (−1.23 to −1.55 from bregma), posterior-middle (−1.67 to −2.03 from bregma) and posterior (−2.15 to −2.45 from bregma) regions of the BLA.

The selected ten genes were targeted in four sequential HCR rounds. The probe sets (Molecular Instruments) were used as follows: *Sema5a, Grik1, Rorb* (Round 1); *Adamts2, Bdnf* (Round 2); *Cdh13, Otof* (Round 3)*; Lypd1, Etv1, Rspo2* (Round 4). The order of rounds was shifted by sections in order to reduce the possibility of loss of RNA after washing steps[74]. In all rounds, the Rnu6 probe was co-applied and used as a nuclear marker. Sections were processed according to the sequential hybridization chain reaction (HCR) protocol as published previously[75]. In brief, sections were fixed in 4% PFA at 4 °C, dehydrated in serial ethanol washes, and treated with RNAscope Protease IV (ACDBio). Sections were rinsed and hybridized overnight at 37 °C with probes targeting the first gene set. Sections were then washed, and probes were amplified with fluorophores (Alexa 405,488, 546, and 647) overnight at room temperature. Next, sections were washed, and autofluorescence was quenched using a Vector TrueVIEW Autofluorescence Quenching kit (Vector labs cat #SP-8400). Slides were cover-slipped with Prolong Gold antifade Mounting Medium (Invitrogen) and allowed to cure at RT for 2 h before imaging. After each round of imaging, coverslips were removed, and sections were washed to remove mounting medium. The probes were then digested with DNase I (Sigma cat #4716728001), and the next probe set was hybridized. Images were acquired with a Leica SP8 confocal microscope and a 20×/0.75 IMM objective (Leica).

**Data analysis for HCR**
**Spatial organization in the BLA**. Firstly, four images from 4 rounds were superimposed by landmark and serial strain registration. Also, cell segmentation (based on Rnu6 expression) and thresholding fluorescence for positive cell per each gene were performed and quantification for cells expressing a given gene was also analyzed by HALO software (Indica Labs). Thresholding required to classify a given gene positive/negative cell was chosen based on visual inspection but double-blind way. Next, we referred to a recent anatomy paper[46] for subregion delineation and coronal section selection of BLA. In order to represent the whole BLA, our dataset constituted four coronal sections (anterior, anterior middle, posterior middle and posterior) and eight subregions (aLA, pLA, amBA, alBA, acBA, pBA, ppBA, ppLA). All subregions except ppLA, ppBA were used in the recent anatomy paper[46]. The citation did not mention the terms ppLA and ppBA, but they analyzed these very posterior parts distinct from other parts of the BLA. Therefore, we named these very posterior parts as 'pp' (Table 1 with abbreviation). To visualize gene expression, positive cells were reconstructed and plotted in dot colorized by gene. Finally, percentage of positive cells for each gene was calculated within individual eight sub-regions. We did not separate multiple gene positive cells. Therefore, the sum of percentage per subregion is more than 100%. Lastly, we compared this percentage across coronal section as well as across subregions in Fig. 2. Pearson's cross-correlation and clustering analysis was computed between BLA subregions and the average of each percentage per gene. The density heatmap was plotted in a color-scale (red = 1 and blue = 0, normalized value by the highest density area (=1)). To map smFISH and snRNA-seq data correlation, we computed pairwise Pearson correlation with hierarchical clustering of 10 marker gene expression across all glutamatergic neurons (snRNA data) and of average percentage of 10 marker gene positive cell across eight sub regions of BLA (smFISH) (Supplementary Fig. 5).

## Table 1 | Alphabetical list of abbreviations

| abbreviation table | |
|---|---|
| **ABRREVIATION** | **DEFFINITION** |
| AB | accessory basal |
| AOD | acousto-optic deflector |
| AAV | adenoassociated viruses |
| ATP | Adenosine triphosphate |
| AMPA | α-am+A1:B72ino-3-hydroxy-5-methyl-4-iso-xazolepropionic acid |
| ANOVA | Analysis of variance |
| ACR | anion channel rhodopsins |
| ACSF | artificial cerebrospinal fluid |
| AC | auditory cortex |
| ASD | autism |
| BA | Basal amygdala |
| B | Basal amygdala |
| BM | Basomedial amygdala |
| BMA | Basomedial amygdala |
| BNST | bed nucleus of the stria terminalis |
| CGRP | Calcitonin gene-relted peptide |
| CGE | Caudal Ganglion Eminence |
| CEA | Central amygdala |
| CCK | cholecystokinin |
| CFC | Conditional Fear conditioning |
| CPP | Conditioned place preference |
| CRH | Corticotrophin-releasing hormone |
| DE | Differential expression |
| DEG | differently expressed gene |
| DIO | Double floxed inverted open reading frame |
| EYFP | Enhanced yellow fluorescent protein |
| EDTA | Ethylenediaminetetraacetic acid |
| GABA | gamma amino-butyric acid |
| GO | Gene ontology |
| GECI | Genetically encoded calcium indicators |
| GRIN | gradient index |
| HCR | Hairpin chain reaction |
| NpHR | Halorhodopsin |
| IL | infralimbic |
| IPSC | inhibitory postsynaptic current |
| ITC | intercalated cells |
| IP | intraperitoneal |
| LA | Lateral amygdala |
| LGE | Lateral ganglion eminence |
| LED | Light emitting diode |
| ME | Medial amygdala |
| NGE | Medial ganglion eminence |
| NGEM | Nanoliterscale Gel Beads-in-emulsion |
| NAC | nucleus accumbens |
| NA | numerical appeture |
| OFT | Open field task |
| PVT | paraventricular nucleus of the thalamus |
| PSTH | peri-stimulus time histogram |
| PBS | phosphate-buffered saline |
| PA | Posterior amygdala |
| PPBA | Posterior-Posterior basal amygdala |
| PPLA | Posterior-Posterior Lateral amygdala |
| PFC | prefrontal cortices |

## Table 1 (continued) | Alphabetical list of abbreviations

| abbreviation table | |
|---|---|
| **ABRREVIATION** | **DEFINITION** |
| PCA | Principal component analysis |
| PKCD | protein kinase C δ |
| ROI | region of interest |
| SCT | secretin |
| SMFISH | Sequential multiplexed (multi-florescent) in situ hybridization |
| SNN | shared nearest neighbor |
| SST | Somatostatin |
| SD | Standard Deviation |
| UMAP | Uniform Manifold Approximation and Projection |

**smFISH PCA clustering analysis.** Data including x,y position of positive cells expressing each gene were imported to a Python workflow in which unsupervised principal component analysis (PCA) were simply customized from the pipeline (EASI-FISH) described before[76]. In brief, Images containing expression patterns of 10 marker-genes were decomposed into principal components (PCs). The eigen-images from the top 4 PCs explained on average $80.8 \pm 5.16\%$ of variance in each sample. PCA of the expression patterns of 10 marker-genes were reconstructed and used to identify spatial patterns in BLA orientation. As the pattern demarcating BLA by each PC component across different samples was homogeneous, we only selected samples with this homogenous pattern of PCA to make populational analysis. Therefore, PC loading values for each gene were averaged by different coronal sections (total 16 PC variance, e.g., anterior PC1, anterior PC2 or anterior-middle PC1, anterior-middle PC2 etc.) and clustered by Pearson's correlation across genes. This analysis was compared with the clusters from percentages of cells positive for 10 marker genes in eight subregions of BLA in a supervised manner.

**Correlation between smFISH and snRNA clusters.** For mapping clusters of snRNAseq data to smFISH signals and corresponding locations, we first aggregated the single nuclei read counts for each cluster for each gene that was used in smFISH (Supplementary Fig. 6). We then correlated (Pearson) the smFISH expression data—normalized z-scores of 10 marker genes in a of radius $50 \, \mu m$[74]—with each cluster expression pattern and assigned each smFISH cell to one of the 11 clusters according to the highest correlation coefficient.

**Stereotaxic surgeries.** Mice were anesthetized for surgery with isoflurane (1.5–2%) and placed in a stereotaxic frame (Kopf Instruments). Body temperature was maintained with a heating pad. A systemic anesthetic (carprofen 5 mg/kg bodyweight) was administered. Mice used in in vitro and in vivo optogenetic experiments were bilaterally injected with 0.4 µl of optogenetic or control virus in the BLA by using the following coordinates calculated with respect to the bregma: −1.8 mm anteroposterior, ± 3.25 mm lateral, −4.75 mm ventral for Lypd1-Cre mice, bregma: −1.5 mm anteroposterior, ± 3.25 mm lateral, −4.8 mm ventral for Rspo2- and Etv-Cre mice. In the same surgery, mice used in optogenetic experiments were bilaterally implanted with optic fibers (200-µm core, 0.5 NA, 1.25-mm ferrule (Thorlabs)) above the BLA ( − 4.6 mm ventral). Implants were secured with cyanoacrylic glue, and the exposed skull was covered with dental acrylic (Paladur). Mice used in in vivo calcium imaging experiments were injected in the right BLA (coordinates as above) with 0.4 µl AAV-GCaMP6f virus. One week later, the microendoscope was implanted. To do so, a 0.8-mm hole was drilled in the skull above the BLA. Debris was removed from the hole, and a sterile 20-gauge needle was slowly lowered into the brain to a depth of −4.8 mm from the cortical surface to clear a path for

the lens. The GRIN lens (GLP-0673; diameter, 0.6 mm; length, -7.3 mm, Inscopix) was slowly lowered into the brain to −4.75 mm from the bregma by using a custom lens holder. The lens was secured in place with glue (Loctite 4305) and dental cement (Paladur). The exposed top of the lens was protected by a covering of a silicone adhesive (Kwik-cast). Approximately four weeks after lens implantation, the mice were assessed for observable GCaMP6 fluorescence. The heads of the mice were fixed, and the top of the lens was cleaned of debris. The miniature microscope (Inscopix) with a baseplate (BLP-2, Inscopix) was positioned above the lens such that GCaMP6 fluorescence and neural dynamics were observed. The mice were anesthetized with isoflurane, and the baseplate was secured with dental cement (Vertise Flow). A baseplate cap (BCP-2, Inscopix) was left in place until imaging experiments. Expression in Etv1-CreER animals was induced by intraperitoneal injections of tamoxifen (150–200 μl, 10 mg/ml, dissolved in 90% cornoil and 10% ethanol) two days after surgery on 4 consecutive days in the modified way as described[77].

**Acute brain slice preparation and electrophysiological recordings.** The animals were anesthetized with isoflurane and decapitated under deep anesthesia. The brain was immediately immersed in an ice-cold cutting solution consisting of NaCl (30 mM), KCl (4.5 mM), MgCl$_2$ (1 mM), NaHCO$_3$ (26 mM), NaH$_2$PO$_4$ (1.2 mM), glucose (10 mM), and sucrose (194 mM), equilibrated with a 95% O$_2$/5% CO$_2$ gas mixture. The brain was sectioned into slices of 280 μm thickness using a Leica VT1000S vibratome and transferred to an artificial cerebrospinal fluid (aCSF) solution containing NaCl (124 mM), KCl (4.5 mM), MgCl$_2$ (1 mM), NaHCO$_3$ (26 mM), NaH$_2$PO$_4$ (1.2 mM), glucose (10 mM), and CaCl$_2$ (2 mM), equilibrated with 95% O$_2$/5% CO$_2$ gas mixture and maintained at 30–32 °C for 1 h before being returned to room temperature.

The brain slices were mounted in a recording chamber and perfused continuously with the aforementioned aCSF solution equilibrated with 95% O2/5% CO$_2$ gas mixture at 30–32 °C. Whole-cell patch-clamp recordings were performed using patch pipettes prepared from filament-containing borosilicate micropipettes with a resistance of 5–7 MΩ. The intracellular solution used for recordings contained potassium gluconate (130 mM), KCl (10 mM), MgCl$_2$ (2 mM), HEPES (10 mM), Na-ATP (2 mM), Na2GTP (0.2 mM) and had an osmolarity of 290 mOsm. The brain slices were visualized using an IR-DIC equipped fluorescence microscope (Olympus BX51) and data were acquired using a MultiClamp 700B amplifier, a Digidata 1550 digitizer, and analyzed using the Clampex 10.3 and Clampfit software from Molecular Devices. The data were sampled at 10 kHz and filtered at 2 kHz.

For optogenetic studies, stimulation of neurons was achieved using a multi-LED array system (CoolLED) connected to the aforementioned Olympus BX51 microscope.

**Behavior paradigms.** All mice were handled and habituated in the behavioral chamber for 4–5 days before experiments. For optogenetic experiments, mice were tethered to the optic-fiber patch cords and habituated to the context for 15 min daily. For calcium imaging experiments, a dummy mini-scope and cable (Inscopix) were fixed on the head of mice and habituated to the context for 20–30 min daily. The behavior arenas were housed inside soundproof chamber equipment with houselights and video cameras (c920 webcam, Logitech).

**Free feeding.** The handling and habituation were a prerequisite for all behavioral experiments. The mice were habituated to the feeding/imaging setup with the miniscope and cable attached for 20 min per day and at least 1 week. On the experimental day, the mice were allowed to explore the cage to identify the food pellet before recording and once the mouse was near the food, recording was started. Mice were given enough food in between the fasting sessions to keep 90% of their normal weight. Food-restricted mice were placed in an empty

home cage (20 cm × 32.5 cm) with a plastic food container fixed to one corner. Food was freely accessible for 10 min per day during 2 days. For optogenetic experiments, mice were continuously photostimulated for 10 min on one day and left with lights off on another day. The light on-off order was pseudo-randomized within a cohort to reduce any effects from the order of photostimulation. After 10 min, the remaining food was weighed. The session was video recorded, and feeding behaviors (e.g., frequency to food container or cumulative time in food container) were also analyzed by EthoVision XT 16.0 video tracking software (Noldus). The recording time for calcium imaging was 15 min. The mice tested with food restriction had a small pellet after the experiment to avoid over-eating on the second day of food restriction. We measured the body weights before/after each experiment and made sure that it was the same on day 1 and 2. If the body weight decreased, we waited more time for the mouse to recover to the same weight as on day 1.

**Social interaction test.** The three-chamber test was performed as previously described[78], but in order to combine with optogenetic and calcium imaging experiments, the door between the chambers was removed. In brief, the novel mice, younger than the test mouse and same gender, were handled for 3 min and then habituated in a wire cage placed in the 3-chamber apparatus for 5–10 min for 4 consecutive days before starting the experiment. The test mouse was located in the center chamber. A wired cup with a novel mouse and an empty cup were introduced into the other two chambers and the sociability test was started. The movement of the test mouse was tracked for 15 min (EthoVision XT 16.0) for calcium imaging. For optogenetic experiments, two days (one day with photostimulation and another day without, but pseudo-randomized order of light on-off epochs within a cohort) were examined and novel mice were changed every day. Sociability was analyzed using cumulative time/frequency in the social zone. Cumulative time in the non-social zone (empty cage) was analyzed for comparison. For each set of experiments, the orientation of the two wired cups containing novel mouse or left empty was counterbalanced.

For several calcium imaging cases, we used a round social arena as described previously[79] as a two-chamber social assay instead of three chambers. In brief, the round-shaped arena (inner diameter: 49 cm, height: 45 cm) was equipped with one 3D-printed transparent bar cage (diameter: 8 cm, height: 10.5 cm) in the center. The inner cage was topped with a cone-shaped 3D-printed roof to prevent the test mouse from climbing up. Inside the cone-shaped roof, a wide-angle (180°) fish-eye lens camera was installed to provide a close-up view of animals' social interactions with high temporal/spatial resolution. Above the arena, a camera at the ceiling was used to track animal's positions and speed with minimal blind spots. Micro-social behaviors such as the exact time point of the start of social interaction or sniffing were measured manually through a wide-angle fish eye camera, as well as automatically tracked by EthoVision XT 16.0 from ceiling camera.

**Contextual fear conditioning (cFC).** Main cFC paradigm was modified from a previous study[33]. On day 1, mice were placed into a contextual fear conditioning chamber (Med Associates) while bilaterally connected to optic fiber cables and received three foot-shocks (0.75 mA for 2 s) at the 198-s, 278-s and 358-s time points. For optogenetic activation experiments, simultaneously with the footshocks, a 10-s, 20-Hz train of 15-ms pulses of 473-nm (10–15 mW) light was used for photostimulation, and constant light of 620 nm (10 mW) was used for photoinhibition. On day 2, mice were connected to optic fiber patch cables and placed in the fear conditioning chamber for 180 s, and neither footshock nor laser light was delivered. Freezing behavior, defined as complete immobility with the exception of breathing, was used as a proxy of fear response. Freezing was automatically quantified using the software ANYmaze 7.2 (Stoelting) as described previously[80].

In brief, the software calculated a "freezing score" depending on the number of pixel changes between frames. If the freezing score fell below an empirically determined threshold for at least 2 s, mice were considered to be freezing. To exclude errors where resting was incorrectly detected as freezing behavior, manually freezing behaviors were verified. Animals were excluded from further analysis if they did not show any freezing behavior upon fear conditioning in a recall session.

**Optogenetic manipulations.** Mice were bilaterally tied to optic-fiber patch cords (Plexon Inc) connected to a 465-nm LED (for ChR2) via Optogenetic LED module (Plexon Inc) and mating sleeve (Thorlabs). Photostimulation was performed using 10 ms, 463-nm light pulses at 20 Hz and 10 mW. Photoinhibition used constant 620-nm light at 10 mW. The LED was triggered, and pulses were controlled PlexBright 4 Channel Optogenetic Controller by with Radiant Software (Plexon Inc).

**Optogenetic conditional place preference (avoidance) test.** Conditioned place preference (CPP) was carried out essentially as previously described[52]. It was conducted in a custom-built arena made of two chambers: rectangular-shaped 2 chambers (45 × 15 cm); one compartment consisted of white walls and a metal floor with circular holes, and the another had red walls and square holes. For the optogenetic experiments, on the pretest day (day 1) optic cable-tethered mice were freely exploring the chambers without light for 10 min after 5 min of habituation. Based on the total time in each chamber, the preferred chamber was identified on that day. For optogenetic activation by ChR2, preference was measured for Lypd1-Cre, but avoidance was measured for Etv1-Cre and Rspo2-Cre mice. Therefore, the preferred chamber was paired with photoactivation for Rspo2-Cre and Etv1-Cre mice, but the non-preferred chamber was paired with photoactivation for Lypd1-Cre mice, for three consecutive conditioning days (day 2–4). However, for optogenetic inhibition by eNpHR 3.0, oppositely, preference was measured for Etv1-Cre, but avoidance was measured for Lypd1-Cre. Therefore, the preferred chamber was paired with photoinhibition for Lypd1-Cre mice, but the non-preferred chamber was paired with photoinhibition for Etv1-Cre mice. During conditioning days mice were constrained in a paired chamber with light for 15 min and another chamber without light for 15 min. On the post-test day (day 5) mice were freely exploring the chambers without light the same as on the pre-test (day 1). The times each animal spent in each chamber and its locomotor activity (distance traveled) were recorded using EthoVision XT 16.0 (Noldus) tracking software. The preference index was calculated by (duration in the paired chamber) −(duration in the non-paired chamber).

**Open field task (OFT).** OFT was carried out essentially as previously described[24]. In brief, four 3-min epochs beginning with a light-off (OFF) baseline epoch, followed by a light-on (ON) illumination epoch, in total a single 12-min session. For analysis, the first light-off and last light-on epochs were excluded in order to avoid novelty or satiation-driven factors.

**In vivo Ca²⁺ imaging of freely moving mice.** $Ca^{2+}$ videos were acquired at 15 frames per second with an automatic exposure length. An optimal LED power was selected for each mouse to optimize the dynamic range of pixel values in the field of view, and the same LED settings were used for each mouse throughout the series of imaging sessions. $Ca^{2+}$ videos were recorded using nVista acquisition software (Inscopix, Palo Alto, CA). To later account for any lag between the onset of behavior and $Ca^{2+}$ movies, a continuous train of TTL pulses was sent from Ethovision XT 16.0 or ANY-maze 7.1 (Stoelting) to nVista acquisition software at 1 Hz and a 50% duty cycle for the duration of the session to synchronize the extracted behavior statistics with calcium traces. The TTL

emission-reception delay is negligible (less than 30 ms), therefore the behavioral statistics time series can be synchronized with calcium traces by the emission/receival time on both devices, using a custom python script. We used the IDPS (Inscopix data processing software, version 1.8.0) for the acquisition of calcium image data, rigid motion correction, automatic selection of neuro somata as the regions of interests (ROIs), and extraction of raw calcium traces by using option, Cnmfe in IDPS and visual inspection with their tracing and morphology. To prevent potential biases resulting from temporal convolution in the calcium traces, we performed spike deconvolution using the OASIS algorithm implemented in Suite2p[81]. The inferred spike trains were used in the following social and food preference experiment analyses.

**Calcium data analysis for feeding and social interaction assays.** The relative distance between the recorded mice and food or other mice is closely related to food consumption and social behavior, respectively. Therefore, we computed this relative distance for each calcium frame recorded in the food consumption or social behavior experiments. This relative distance was then normalized by the radius of the experiment chamber size.

To inspect the correlation between neuron firing rate and the relative distance to food or other animals, we divided the relative distances into 31 bins, and computed the averaged spike firing rate of the frames whose relative distances fell into the same distance bin. The preferred relative distance of each neuron was determined as the distance bin with the highest averaged firing rate. The neurons were classified into difference valence-correlated categories based on their preferred distance to food/other animals.

**Permutation test.** In order to confirm that neuronal activities related to specific contexts (food and social assay) beyond chance level (null hypothesis), we rotary shuffled the inferred spike train for each neuron with a random time offset for 1000 times. For each shuffled spike train, we computed the null distance distribution by calculating the averaged firing rate for each distance bin. We used Benjamini−Hochberg procedure[82] to control for a false discovery rate at 5%. Therefore, if the maximum average firing rate of the distance distribution of a neuron was higher than 95% of the null distribution, the neuron was considered significantly tuned to distance to food/social object (Fig. 4M, N, Fig. 5G, H). To determine if the differences in the percentage of pro/anti-food/social neurons between Lypd1, Etv1 and Rspo2 neuron population were significant, we pooled the neurons from the three populations. For each neural population, we randomly selected N neurons from the pool distribution and computed the percentage of pro/anti food/social neurons for 1000 times to obtain a null percentage distribution (N equals the number of neurons for the testing neural population). We then compared the percentage of the testing population with the two tails of the null percentage distribution and determined the significance at the 2.5% significance level (Figs. 4O and 5I).

**Fear conditioning calcium data analysis.** The freezing behaviors are detected automatically by ANY-maze 7.1 (Stoelting) with the 2-second minimum duration. To determine the correlation between neural activities and foot shock/freezing behavior in the fear conditioning experiment, we computed the score for foot shock as follows:

$$Foot\ shock\ response\ score = (F_{during\ shock\ on} - F_{off\ before\ shock})/(F_{during\ shock\ on} + F_{off\ before\ shock})$$

Similarly, the freezing score was computed as

$$Freezing\ score = (F_{freezing} - F_{non-freezing})/(F_{freezing} + F_{non-freezing})$$

$F_{shock\ on}$ and $F_{freezing}$ are the averaged firing rates in the 2 s before and during the onset of foot shock or freezing events, respectively.

The neurons positively correlated with foot shock events might be involved in the negative valence event representation. To investigate this, we computed the percentage of foot-shock correlated neurons in the fear-acquisition session and the freezing frequency in the fear retrieval session for each mouse. We only included mice having neuronal data on both Day 1 and 2. One (Etv1-Cre) and two (Lypd1-Cre) mice did not have good quality-neuronal data either on Day1 or Day 2 after preprocessing. These mice were excluded from neuronal data analysis. We performed the linear regression on these two statistics for quantitative descriptions of the relationships between these two statistics.

### Classification of footshock responsive neurons

In order to classify positive footshock responsive neurons (pro-footshock) or negative footshock responsive neurons (anti-footshock), we rotary shuffled the inferred spike train for each neuron with a random time offset for 1000 times. Then we computed the mean response score to fear stimulus for each shuffled spike train in the same way as described above, in order to obtain a null response score distribution. The neuron whose mean response score was higher than the top 2.5% of the null distribution was considered a pro-footshock neuron whose activity was positively correlated with a footshock event. Vice versa, a neuron with a response score lower than the bottom 2.5% of the null distribution was considered an anti-footshock neuron.

### Longitudinal tracking of individual neurons for in vivo calcium imaging

To monitor the same neurons for both social interaction and footshock responses, we performed the social interaction test in the fear conditioning chamber. Calcium traces were recorded during an initial period in which the experimental mouse was alone. Then, a wired container with a novel mouse was introduced to the chamber for a "social interaction period". The container was taken out and five consecutive footshocks were delivered (Supplementary Fig. 10A,B). Since the fear conditioning chamber was smaller than the chamber previously used for social interactions, we scored the social response score (SoRC) differently. We extracted the animals' head directions and body center positions using the software ANYmaze 8.2 (Stoelting). Animals were considered to be engaged in social interactions when their heads were oriented towards the wired container and their distance to the center of the container was less than 10 cm. The auto-identification of social interactions using this criterion produced similar results as manual identification (data not shown; Pearson correlation = 0.809). For quantification, we used spike detection to deconvolve calcium traces to firing rates over time and then calculated the average firing rates for the social-interaction phases and non-social phases ($F_{social}$ and $F_{non-social}$, respectively) for each neuron. The social response score (SoRC) was calculated as below:

$$Social\ response\ score = \frac{F_{social} - F_{non-social}}{F_{social} + F_{non-social}}$$

Neurons with a positive SoRC were termed 'social ON' neurons, while neurons with a negative SoRC were termed 'social OFF' neurons. The shock response score (SRC) was calculated as described in Fig. 5 and the method section. However, we did not classify the footshock responsive neurons using the permutation test but rather termed neurons with positive SRC 'shock ON' neurons and those with negative SRC 'shock OFF' neurons.

### Immunohistochemistry

For recovery of neurobiotin-filled neurons after whole-cell recordings, acute brain slices were fixed in 4% PFA at room temperature for 30–45 min. Fixed slices were kept in 0.1 M PB (80 mM $Na_2HPO_4$ and 20 mM $NaH_2PO_4$) until being processed for immunohistochemistry. Slices were then washed in 0.1 M PB and incubated with fluorophore-conjugated streptavidin (1:2000) (Jackson) diluted in 0.05 M TBS with 0.5% Triton X-100 overnight. The next day, slices were washed in 0.1 M PB and mounted with RapiClear (SunJin Lab Co). Slices were imaged 1 d later. Coronal brain sections were permeabilized for 30 min at RT with 0.5% Triton X-100 (#66831, Carl Roth) in PBS and blocked for 30 min at RT with 0.2% BSA (#A7030, Sigma-Aldrich) and 5% (w/v) donkey serum (#017-000-121, Jackson ImmunoResearch) in PBS. Sections were incubated with primary antibodies in 0.2% (w/v) BSA and 0.25% Triton X-100 in PBS at 4 °C overnight. The following primary antibodies were used: 1:100 rabbit anti-Slc17a7 (48-2400, Invitrogen). Sections were washed three times for 10 min with PBS and incubated for two hours at 4 °C with secondary antibodies diluted 1:400 in 0.2% (w/v) BSA and 0.25% Triton X-100 in PBS. The following secondary antibodies were used: donkey anti-rabbit Alexa Fluor 488 (anti-rabbit, 711-545-152, Jackson ImmunoResearch). Sections were washed two times for 10 min with PBS and incubated with DAPI (1:2000) (Sigma-Aldrich) in PBS. After a 15-min wash in PBS, sections were mounted using Fluorescent Mounting Medium (#S3023, Dako).

### Histology

Animals were anesthetized IP with a mix of ketamine/xylazine (100 mg/kg and 16 mg/kg, respectively) (Medistar and Serumwerk) and transcardially perfused with ice-cold phosphate-buffered saline (PBS), followed by 4% PFA (1004005, Merck) (w/v) in PBS. Brains were postfixed at 4 °C in 4% PFA (w/v) in PBS overnight, embedded in 4% agarose (#01280, Biomol) (w/v) in PBS, and sliced (50–100 μm) using a Vibratome (VT1000S−Leica). Epifluorescence images were obtained with an upright epifluorescence microscope (Zeiss) with 10x or ×5/0.3 objectives (Zeiss). To acquire Fluorescence z-stack images, a Leica SP8 confocal microscope equipped with a 20x/0.75 IMM objective (Leica) was used. For full views of the brain slices, a tile scan and automated mosaic merge functions of Leica LAS AF software were used. Images were minimally processed with ImageJ software (NIH) to adjust for brightness and contrast for optimal representation of the data, always keeping the same levels of modifications between control and treated animals.

### Data analysis

Data and statistical analyses were performed using Prism v5 (Graph-Pad, USA) and Excel 2016 (Microsoft, USA). Clampfit software (Molecular Devices, USA) was used to analyze electrophysiological recordings and all statistics are indicated in the figure legends. T-tests or Ordinary one-way ANOVA with Tukey's multiple comparisons test or two-way ANOVA with Bonferroni post-hoc tests were used for individual comparisons of normally distributed data. Normality was assessed using D'Agostino & Pearson normality test. When normality was not assumed Kolmogorov–Smirnov test and Wilcoxon signed-rank test were performed for individual comparisons. $P$-values represent $*p < 0.05$; $**p < 0.01$; $***p < 0.001$. All data were represented as the mean ± SEM or STD. All sample sizes and definitions are provided in the figure legends. After the conclusion of experiments, virus expression and implant placement were verified. Mice with very low or null virus expression were excluded from the analysis.

### Reporting summary

Further information on research design is available in the Nature Portfolio Reporting Summary linked to this article.

### Data availability

The raw data for electrophysiology (Fig. 4), Calcium imaging (Figs. 4−5) and optogenetic (Figs. 6−7) have been deposited in the google drive: https://drive.google.com/drive/folders/166KtcgxC-

Ad27MZH-lidATxEQKgC0BMf. Raw and processed snRNAseq data (Fig. 1) are available at GEO (accession number GSE244860). Source data are provided as a Source Data file. Source data are provided with this paper.

## Code availability

Custom-written code for calcium imaging (Figs. 4–5) is publicly available in a GitHub repository at https://github.com/limserenahansol/1p_BLA_sync_permutation_social_valence or at Zenodo: https://doi.org/10.5281/zenodo.11995740.

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

## Acknowledgements

The authors thank Pilar Alcalá for her support in snRNA seq preparation, Yuhan Wang (Janelia Research Campus) for her input in smFISH analysis, Jan Gründemann (DZNE) and Christian Mayer (MPIBI) for their scientific advice during the entire project, Mark Hübener (MPIBI) for reviewing the manuscript. This study was supported by the Max-Planck Society and the European Research Council under the European Union's Horizon 2020 research and innovation programme (no. 885192, BrainRedesign, to R.K.).

## Author contributions

H.L. and R.K. conceptualized and designed the study. H.L. conducted and analyzed most experiments and data. Y.Z. supported constructing calcium-imaging analysis pipeline. C.P. performed electrophysiology experiments. JM supported the preparation of smFISH and behavioral experiments. T.S. contributed to snRNAseq data analysis. H.L. and R.K.

wrote the paper with input from all authors. R.K. supervised and provided funding.

## Funding

## Competing interests

The authors declare no competing interests.

## Additional information

**Peer review information** : *Nature Communications* thanks the anonymous reviewers for their contribution to the peer review of this work. A peer review file is available.

