## [Peer Review File · Nature Communications]

Genetically- and spatially-defined basolateral amygdala neurons control food consumption and social interactionEditorial Note: This manuscript has been previously reviewed at another journal that is not operating a transparent peer review scheme. This document only contains reviewer comments and rebuttal letters for versions considered at *Nature Communications*.

REVIEWER COMMENTS

Reviewer #1 (Remarks to the Author):

The authors discussed extensively all the points I have raised, and I am generally satisfied with their response.

I have just some minor comments:

- Answer to question 4c reads "Lypd1-expressing cells showed little overlap with Etv1-...". Further in question 4f, the author's comment reads "...the extent of colocalization...". Are the terms "overlap", "colocalization" and "occupancy" (Ext. fig 7) all used to describe the anatomical position of the cells rather than the fact that distinct molecular markers are expressed in the same cell (i.e. co-expression)? If so, and to avoid misunderstandings, I would opt for the use of "colocalization (within an area)", rather than "overlap", when referring to the anatomical position of a cell.
- Point 7 and 8. One of the main points I have raised is relative to the fact that the authors used the approach to food as a proxy for eating (rather than just measuring eating). However, the approach might or might not result in food ingestion. The authors now show that in their paradigm, when the mouse approaches the food area (within 5cm) is very likely to ingest food (point 7). This explanation is reasonable to me. However, if the authors want to use the term "feeding" rather than "approach to food" it should be reported in the main text that "feeding", in their paradigm, is not just the act of eating (ingesting) but rather includes multiple phases: 1) the approach to the food (as a main parameter), 2) subsequent lifting/handling of the pellet, and 3) ingesting the pellet (point 8).
- Point 12. I feel it is still not clear why using 2 different arenas (Fig. 5 and Fig. 7) to measure the same behavior (socialization).
- Point 14. The authors added, as I asked, experiments relative to the valence of the photoinhibition of Lypd1 and Etv1 neurons. These new data strengthen their conclusions regarding the involvement of Lypd1 in the regulation of feeding (inhibition of Lypd1 has negative valence and in line with this, when tested for feeding, the mice eat less). The data about photoactivation/photoinhibition of Etv1 are less intuitive. Photoactivation of Etv1 has a negative valence, while photoinhibition induces place preference. However, photoactivation promotes socialization (generally rewarding), while photoinhibition reduces socialization. These results should be interpreted keeping in mind that Etv1 contains more than one population (Ext. Fig. 10, one responding to shock, one to socialization, one to both) and that optogenetic manipulation will affect the activity of all cells together. The authors should discuss these aspects (the valence of the manipulations in CPP tests and how they relate to behavioral changes in feeding, freezing, and social behavior) more in detail.

Reviewer #3 (Remarks to the Author):

The authors have satisfactorily addressed the majority of my comments with the revised version of the manuscript, but there remains one concern outlined below that must be addressed before the associated manuscript should be considered for publication. There is a key set of characterization

experiments that are needed to ensure that the transgenic lines are appropriate for the experiments and interpretations; without this, the manuscript runs the risk of misinterpreting manipulated cells and the sources of behavioural phenotypes.

Major Comment:

The authors' response did not address my point regarding an admitted lack of characterization of the viral model: "In our hands, viral reporting usually follows well the spatial pattern of genetic reporters but is more variable in terms of the number of cells labeled. We therefore felt that quantification of genetic reporter expression would suffice." Without this characterization, this makes interpreting the results of the optogenetics experiments difficult. Firstly, reporter crosses can vary substantially from viral reporting due to nature of reporting (crosses permanently report expression of marker gene at any time; viruses report expression roughly only at the time of injection and are shaped by the type and properties of the virus). With no data that validates that *Lypd1*, *Etv1* or *Rspo2* cells were successfully targeted for optogenetic manipulation, and that these cells are restricted to the BLA, we cannot be sure that these are the cell types that are responsible for the effects that the authors attribute to them. High enough viral titre can always induce off-target expression of opsins, random insertions (like those used in the *Lypd1*-cre line from GENSAT BACs) can drive expression in off target cells, and viral spillover outside of the BLA can also contribute to off-target effects. The cell-type specificity, both to the "ground truth" marker gene of interest as well as restriction of expression to the BLA, must be properly quantified to verify that each of the AAV-mediated opsin expressions are faithful to the cell types that are suggested to have been targeted for optogenetic manipulation.

Minor Comment:

Regarding the request for clarification about how hundreds of trials were achieved in Figure 5B, the authors did not clarify how many animals and how many individual trials were performed, which could have been provided in a single line: x animals * x neurons per experiment * x trials * x shocks per trial. As such, the methodology here remains opaque, clouding interpretation of the data.

Response to reviewers' comments

Reviewer #1:

Comments for the Author:

Lim et al. carried out single-nuclei RNA sequencing and smFISH of the basolateral amygdala and identified 11 glutamatergic and 10 GABAergic clusters of neurons. They focus their attention on the glutamatergic neurons and provide compelling evidence for the notion that molecularly encoded cell-types direct specific functions. Of interest, they identify a population of *Lypd1* that participates in feeding behavior and a mixed population expressing *Etv1*, regulating social behavior and contextual fear conditioning.

I commend the authors for the variety and the complementarity of their approaches. This work will be of great help to the field and to those who investigate the basolateral amygdala. I do have, however, some comments. Given the growing number of single-cell or nuclei datasets related to the amygdala, I feel it is very important to acknowledge and discuss this data in relation to previously published datasets. Although the authors partly address this aspect, I believe that differences and analogies with the recently published dataset of Hochgerner et al need to be more thoroughly discussed. Furthermore, some of the data presented, in particular those relative to the feeding intake of mice, require some explanation. Considering that the finding that a genetically encoded population of BLA neurons regulates feeding is quite novel (at least to be best of my knowledge), and is quite central in the discussion, it is of pivotal importance that these concerns are carefully addressed.

Main comments:

- 1) Line 99: "a second population showed mixed selectivity by responding to aversive and social cues": considering that defensive behavior and social behavior are somehow very different behaviors, can it be that this population of cells (i.e. *Etv1*) contain at least 2 subpopulations? A way to find out if this is the case is via their imaging experiments: the authors tested whether *Etv1* neurons are activated during feeding-related behaviors, during contextual fear conditioning, and during social interactions with single cell resolution. Can the authors show if the neurons that are activated during defensive behavior and social behavior are the same? If not, this would inform whether *Etv1* neurons are more heterogeneous than transcriptomics data suggest or confirm that there exists a cell type encoding two, at a glance quite distant, behaviors and emotional states.

We agree with this reviewer that these two alternative possibilities exist and conducted an additional calcium imaging experiment (with new cohorts of mice) in which *Etv1* and *Lypd1* neurons were imaged at single-cell resolution, while the mice were subjected to social cues and footshocks in the same session. The new results confirm that the BLA^{*Etv1*} population contains a large fraction of neurons that are activated during social and defensive behavior. The data were added to the manuscript (Line 375) and in Extended data Fig. 10.

To monitor the same neurons for both social interaction and footshock responses, we performed the social interaction test in the fear conditioning chamber. Calcium traces were recorded during an initial period in which the experimental mouse was alone. Then, a wired container with a novel mouse was introduced to the chamber for a "social interaction period". The container was taken out and five consecutive footshocks were delivered (Extended data Fig. 10A,B). Example traces of neurons active during social interactions and/or footshocks are shown in Extended data Fig. 10C. Since the fear conditioning chamber was smaller than the chamber previously used for social interactions, we scored the social response score (SoRC) differently (also described in the methods, line 1019). We extracted the animals' head directions and body center positions using the software ANYmaze 8.2 (Stoelting). Animals were considered to be engaged in social interactions when their heads were oriented towards the wired container and their distance to the center of the container was less than 10 cm. The auto-identification of social interactions using this criterion produced similar results as manual identification (data not shown; Pearson correlation = 0.809). For quantification, we used spike detection to deconvolve calcium traces to firing rates over time and then calculated the average firing rates for the social-interaction phases and non-social phases (F_{social} and $F_{\text{non-social}}$, respectively) for each neuron. From this quantification, we calculated the SoRC. Neurons with a positive SoRC were termed 'social ON' neurons, while neurons with a negative SoRC were termed 'social OFF' neurons. The shock response score (SRC) was calculated as described in main Figure 5 and the method section. However, we did not classify the footshock responsive neurons using the permutation test, but rather termed neurons with a positive SRC 'shock ON' neurons and those with a negative SRC 'shock OFF' neurons.

The quantification revealed that the largest fraction of BLA^{Etv1} neurons (44%) were both ‘social ON’ and ‘shock ON’ neurons and outnumbered the fractions that were either ‘social ON’ or ‘shock ON’ neurons (16 and 22%, respectively; Extended data Fig. 10D, n=481 cells from 4 animals). Linear regression analysis revealed a significant positive correlation in the SRCs and SoRCs in BLA^{Etv1} neurons ($p= 0.01$) although the data showed high variability with low R^2 value (Extended data Fig. 10E). In comparison, the ‘social ON’ and ‘shock ON’ fraction of BLA^{Lypd1} neurons was rather small (21%, Extended data Fig. 10F, n=422 cells from 3 animals), consistent with the results on social interactions in main Figure 5. In that experiment, we found that the fraction of pro-social BLA^{Lypd1} neurons was significantly smaller compared to shuffled data. Moreover, linear regression analysis revealed a lack of positive correlation in the SRCs and SoRCs in BLA^{Lypd1} neurons ($p= 0.41$; Extended data Fig. 10G). These results confirm that the BLA^{Etv1} population contains a large fraction of neurons that are activated during social and defensive behavior.

- 2) In the methods section it reads that the mice used in the study were juvenile (postnatal days 10 and 21) or adult mice (8 weeks or more). I think which was used for what experiment is not clearly specified. I guess the juvenile mice were used for single nuclei sequencing, is it the case? Why 10 or 21 days, what is the reason for this age gap? And was the validation via smFISH carried out in adult mice or juvenile mice? This information, and information regarding the sequencing methods, the number of mice used, their age, and their sex should be readily available to the reader, possibly in the main text, e.g. around Lines 110-111.

Sorry for this mistake. The juvenile data was for another project and was not included in this manuscript. The method paragraph was revised accordingly. The information requested by the reviewer was provided (Line 111).

- 3) The findings of Hochgerner et al is mentioned on line 123, but not in the discussion. In their article, now published in Nature Neuroscience (2023), Hochgerner and colleagues carry out a comprehensive molecular analysis of the central and basolateral amygdala (and nearby areas) and identify known and new genes responsive to fear learning and cell types involved in fear and memory retrieval. I appreciated Lim and colleagues discussing how their findings compare to O'Leary's, but how do their findings match or differ from Hochgerner's?

We agree and compared our data with the recent whole amygdala transcriptomics study (Hochgerner et al 2023). We extracted the data concerning all of the excitatory BLA clusters from this study and integrated it with our data. We found that several of the main clusters matched well with our clusters (Extended data Fig. 14A, B). For example, our BA cluster 1 (*Rspo2/Etv1/Adamts2/Bdnf*) showed a near-perfect match with Hochgerner's VGLUT1-2 cluster from anterior BA (Extended data Fig. 14C, D). Our clusters from the LA (cluster 2, *Sema5a/Otof/Lypd1/Cdh13* and cluster 8, *Rorb/Lypd1*) showed similarity to Hochgerner's VGLUT1-28 and VGLUT1-29 clusters from LA (Extended data Fig. 14C, E). A comparison of genes matching in both datasets revealed that the pattern of gene expression was distinct between the BA and LA clusters supporting our idea that these subregions contain genetically-defined subpopulations (Extended data Fig. 14D, E). Some of our most anterior and most posterior located clusters (our clusters 5 and 7) remained unassigned in the Hochgerner subset. Since part of the spatial distribution in the Hochgerner study was inferred by correlation with the Allen Mouse Brain Atlas, we speculate that these clusters were wrongly assigned to non-BLA amygdala clusters. We have added this analysis to the discussion (line 507).

Regarding the overall depth of analysis, we found that our data included more excitatory BLA neurons than the Hochgerner study ($n=1960$ versus $n=1185$, see table below) and our dataset contained a larger number of unique cells ($n=697$ versus $n=82$).

	VGLUT1-1	VGLUT1-1	VGLUT1-1	VGLUT1-2	VGLUT1-2	VGLUT1-2	VGLUT1-3	VGLUT1-3	VGLUT1-6	VGLUT2-2	Cells unmatched to HG clusters in our data	Cells matched to HG clusters in our data
Etv1 (am/alBA)	0	1	0	2	37	0	0	0	163	0	45	248
Etv1/Lydp1 (acBA)	0	0	63	1	0	1	0	0	3	0	10	78
Grik1 (acBA,pLA)	0	0	0	0	3	0	0	0	5	0	41	49
Otof (pBA)	8	42	4	6	0	2	66	1	19	0	106	254
Otof/Lydp1/Cdh13 (pLA,pBA)	1	5	55	2	13	50	1	0	9	0	105	241
Rorb (aLA)	0	4	0	4	7	2	3	0	8	0	119	147
Rorb/Lydp1 (aLA,pLA)	0	0	0	0	129	0	0	0	0	0	12	141
Rspo2 (am/alBA)	0	0	0	1	0	0	0	0	11	0	57	69
Rspo2/Etv1/Adams2/Bdnf (aB.	0	2	0	196	4	1	7	0	5	0	52	267
Sema5a (pLA)	0	0	0	0	1	4	0	0	1	71	131	208
Sema5a/Otof/Lydp1/Cdh13 (p	0	0	0	0	110	126	3	0	0	0	19	258
											697	1960

all cell numbers in our BLA glutamatergic clusters.

	Etv1 (am/	Etv1/Lydp1	Otof (pBA	Otof/Lydp1	Rorb/Lydp1	Rorb/Lydp1	Rspo2/Etv1	Sema5a (p	Sema5a/C	Cells unmatched to our clusters inHG	Cells matched to our clusters in Hochgerner data
VGLUT1-13-Dcn_C1qI2	0	40	12	5	0	0	0	0	0	44	101
VGLUT1-14-Cartpt_Fam46a	0	5	5	1	0	0	0	0	0	5	16
VGLUT1-15-Fmo1_Rxfp3	0	70	3	15	0	0	0	0	4	12	104
VGLUT1-2-Rspo2_Sema3e	6	1	1	0	0	362	0	0	0	1	371
VGLUT1-28-Grp_Cpne8	17	1	0	1	150	3	0	86	0	4	262
VGLUT1-29-St8sia2_Cald1	0	0	0	2	0	0	0	19	0	1	22
VGLUT1-3-Sema5a_Dcn	0	1	25	0	1	38	0	10	0	4	79
VGLUT1-32-Mid1_Cdh22	1	0	2	9	0	0	0	7	0	2	21
VGLUT1-6-Thrsp_Lamp5	180	2	0	11	0	1	0	3	0	3	200
VGLUT2-24-Cdkn1c_Penk	0	0	0	0	0	0	3	0	0	6	9
										82	1185

all cell number of HG BLA cluster

Table 2: unmatched (unassigned) or matched cell numbers per excitatory BLA cluster from our study (top) and from Hochgerner's subset (bottom)

4) Lines 192-201: Extended data Figure 5A and 5B do not show smFISH data. Can the authors show it? In addition, "Rorb" is shown in their plots but it is the only gene not acknowledged in the main text, in the paragraph relative to these figures. Is there a specific reason for this?

Extended data Figure 5 was generated from the main Figures 1 and 2 as mentioned in the legend of Extended data Figure 5. We have rephrased this section in the main text (Line 196) and in the legend and mentioned the Rorb data in the main text (Lines 130, 140, and 161).

5) Extended data fig. 7A-C: the authors analyze the expression of Rspo2, Etv1 and Lydp1 with smFISH. I have more than one comment regarding this:

a. Figure 2A-C is hard to read partly because Rspo2 expression (in blue) is virtually impossible to see, both on pdf on-screen and printed.

b. Panel B, the legend reads that Rspo2 is in blue. I can see the blue signal in the last 2 panels (starting from the left) of Fig. 7B, but is the blue channel present in the first 2 panels (from the left) of Fig. 7B?

Thank you for the helpful comments. We reformatted Extended data Fig. 7 and removed the old panel B. We only used two genes in pairwise comparisons for better visualization and always used green and red colors (in addition to Rnu6 (blue)). Additionally, we added higher magnification images for better visualization of co-expression. Also, our criteria for double- and single positive or negative cells have been explained in the legend of Extended data Fig.7 and the Methods. In brief, if the number of puncta in the segmented cell was higher than the average number of puncta per cell in the whole BLA section, the cell was assigned positive. This was subsequently confirmed by visual inspection by two persons blind to the identity of the probe.

c. Etv1 and Rspo2 co-expression, which according to the text is less than 20% (Line 244) seems way more than 20% if I were to judge from Fig. 2B and virtually completely separated judging from Fig. 2C. Why is that?

We assume the reviewer misread the sentence. It was not about Etv1 and Rspo2 co-expression, but about how much Lypd1-expressing cells overlapped with either Etv1- or Rspo2-expressing cells (Line 246: “*Lypd1*-expressing cells showed little overlap with *Etv1*- or *Rspo2*-expressing cells (typically less than 20% overlap)”).

d. The borders of the BLA appear poorly drawn and in some cases, like Fig. B (third pic from the left) seem to have been drawn twice. Can the authors comment on this and according to what criteria were these borders traced?

Thanks for spotting this error. The images from FISH are superimposed on the reference atlas (Hintiryan et al., 2021) by using the HALO software and tissue landmarks. The borders of the BLA are then drawn based on this superimposition. When the image was transferred to the publication format the line was doubled. The error was corrected in Extended data Fig. 7.

e. Figure 7C is large and would benefit from more letters, to navigate it. Further, the picture just below the letter “C” seems to have the same border as B but the staining appears different. The authors should clarify if C and B share some of the pictures and report this in the legend.

Thanks, we have improved the structure of the figure as requested.

f. Finally, the quantifications often show a lot of variability (to make an example, the graph in the line in the middle is quite extreme, with 2 mice at 70% and one mouse at 20% of “Lypd only %”), which precludes firm conclusions. The authors should add more animals to these experiments.

The most relevant information in this figure is the extent of colocalization between the three cell populations. We found that colocalization of Lypd1 cells with either Etv1- or Rspo2-expressing cells was low (typically less than 20%), whereas colocalization of Etv1- and Rspo2-expressing cells was much higher (between 30 and nearly 60%). The variance of colocalization was rather low and we therefore would argue that this data is sufficient to make that point. We would find it hard to justify adding more animals to these experiments.

6) Extended data fig. 8A:

a. The label “Cre/Etv1” to me means the proportion of Cre+ cells expressing Etv1 on the total Cre+ cells. Viceversa, the label “Etv1/Cre”, indicates, to me, the proportion of Etv1 cells expressing Cre. From the figure legends, it seems that the authors meant the exact opposite.

What is the right way to read these data? Regardless, from B it is clear that at least half of the Etv1 cells (red signal) do not express Cre (green signal). This seems incongruent with the author's quantification. Please explain.

We agree with the reviewer that the labeling of this panel was ambiguous. We have confirmed that the quantification was correct and have reformatted this figure with more representative images as well as high-magnification examples. We also changed the labeling to Tom/Etv1 and Etv1/Tom in the figure and legend.

An example of how we quantified mFISH and Tom expression was added to Extended data Fig.8 E. The white-filled arrows indicate cells positive for both Tom and endogenous mRNA signal, empty arrows mark cells positive only for endogenous mRNA signal, asterisks indicate cells that are not considered for quantification due to lack of a nuclear signal (Dapi).

b. In B, the co-localization of Cre and Etv1 is hard to tell given the low magnification used to take the picture, and insets would help.

We added high-magnification insets.

c. Some pictures have a sort of black frame, which I would remove, as it is not part of the original images, I reckon.

We removed the black frame.

d. One of the images in B seems to have 2 scale bars.

Thanks, this was fixed.

e. In C, it is very hard to identify double positive Cre and Lypd1 cells, due to the fact that Lypd1 signal is in blue, which I would change to red.

Thanks for the suggestion, we changed the color to red.

f. I would also recommend showing data points instead of error bars, for a reason of clarity.

We changed to graphs with data points.

g. Lastly, the label “Cre” is confusing, as this is tomato expression, under the control of Cre, therefore it should read “Tomato” or “tom”.

We changed the label to “tom”.

7) Line 277-289: the authors write: “We quantified mouse feeding behavior according to their approach behavior towards food rather than food consumption because in previous work on central amygdala neurons, the presence of food correlated better with neuron activity than food consumption”. Although their neurons are connected via circuits, the CeA and the BLA are two different brain areas. I, therefore, do not understand the reasoning of only quantify food “approach behavior” and not actual food consumption behavior. I would like the authors to show data acquired via this experiment relative to the actual food consumption, i.e. when the mouse picks up and starts to eat the food, as well as neuronal activity during the feeding bout (i.e. eating). Given that Lypd1 neurons are deemed to have positive valance, I would expect these neurons to be activated during eating.

When using a top view with a single video camera, it was very difficult to precisely observe food pick up and eating every time the mouse approached the food. Manual scoring by different observers agreed that within a distance of 5cm around the food, the mouse was very likely to consume food (in 9/10 cases). That’s why we decided to correlate the calcium data with approach behavior. We do agree that correlating with food pick up and eating would have been more accurate, but this would have required an upgrade of the setup and a repetition of the entire experiment. We have therefore not repeated this experiment for the revision of this manuscript. We have added sentence in the text to better explain the reasons for using approach behavior in our analysis.

Additionally, here we also analyzed the Pearson Correlation value between “in 5cm mouse detection” and “manual scoring (feeding)”. The correlation coefficient is **0.8851** which means highly correlated. Therefore we can rely on our distance-based analysis for feeding behavior and below is an example of how manual scoring (feeding) and body point in 5cm detection are well matched (X axis= time, Yaxis= 0 or 1 (0 means No (no feeding or outside 5cm), 1 means Yes (feeding_or in 5cm)) as seen below: (Blue line is manual scoring for feeding and Red line for automatic detection by distance 5cm. both are highly synchronized). We added more explanation the reason for using approach behavior in Line 287. Also, the file is in source data.

8) Line 287-288: The pro-food area extends 5 cm from the location of the food. Can the author clarify what kind of behavior is classified as “feeding behavior”? Is the actual eating (see point 7) included?

Feeding behavior includes preparing and grabbing the food pellet and eating it.

9) The method session “free feeding” does not mention habituation. Was habituation to the setup carried out for feeding, in a similar way as done for the social tests? This is particularly important for feeding studies, as mice introduced to a new environment might experience anxiety and/or be driven to explore the new space, before engaging in eating, even if hungry. In fact, I noticed that in Figure 6C,D some experimental (i.e. Chr2) mice with light off and some control mice (i.e. YFP) with either light “on” or “off” (therefore no stimulation of *Lypd1* neurons ongoing) did not eat much (less than 0.1g seems rather low, considering that these mice are hungry), with some (C, *Etv1* panel, YFP group) eating 0 grams in a 10-minute session, despite being food-restricted for 20h. This may suggest that the paradigm has some important flaws. I would like the authors to comment on this.

Thanks for these comments. Habituation was not mentioned for the free-feeding experiments, but handling and habituation were a prerequisite for all behavioral experiments. The mice were habituated to the feeding/imaging setup with the miniscope and cable attached for 20 min per day and at least 1 week. On the experimental day, the mice were allowed to explore the cage to identify the food pellet before recording and once the mouse was near the food, recording was started. Mice were given enough food in between the fasting sessions to keep 90% of their normal weight. This may be the reason why some mice did not eat a lot of food during a 10 min interval. We added this detail in the Methods (Line 860),

On a similar note, I would like to read more info about how these experiments were carried out. Were the mice food restricted, then tested for 10 minutes, then satiated at the end of the test? Or were left without food for another 20h, right after the first day of testing? In my experience, mice that underwent food restriction should be satiated at the end of the experiment and rested for 24h or 48h with food ad libitum, before food restricting them and testing them again. This to avoid potentially confounding effects on food intake measures, due to over-eating on the second day of testing (if no food is introduced after the first day of testing, increasing hunger on the second day) or “under-eating” on the second day of testing (if food is reintroduced after the first testing and mice intake more calories that they need, resulting in reduced hunger the second day of testing).

Thank you for these comments. The mice tested with food restriction had a small pellet after the experiment to avoid over-eating on the second day of food restriction. We measured the body weights before and after each experiment and

made sure that it was the same on day 1 and 2. If the body weight decreased, we waited more time for the mouse to recover to the same weight as on day 1. We added this detail in the Methods (Line 872).

10) Line 303, “shuffled data”. How did the authors “randomly selected N neurons” and why not use all neurons instead of a number equal the number of the neurons in the tested population? Shouldn’t one expect values around 50%, in the shuffled group? Lastly, I think it would be more informative to compare the % of significant neurons across the 3 populations rather than to the shuffled values.

We thank the reviewer for these questions. This method may need a better explanation. We rephrased the methods part and main text. In brief, we pooled all neurons from the 3 populations and randomly selected N neurons such that by iterative shuffling the N number is not fixed, but randomly selected. This creates a null distribution which hypothesizes that the variance in the data can be explained by population-irrelevant factors. Using this empirical null distribution, we constructed a non-parametric permutation test to analyze how significant the variance is to reject the null hypothesis without making other assumptions to avoid any potential analysis biases (for example, the normality assumption which expects the values to be centered at 50%). We rephrased this part in the main text (Line 303).

11) For social behaviors (and in general a comment valid of all mice in the manuscript): are the mice used in the experiment males or females?

Thank you for the comment. We will explain this more carefully, although it was mentioned in the Methods. We used both sexes but we always put a non-familiar, same-sex, but younger mouse into the wired container as a social affective paradigm. We added this information in the main text (Line 355).

12) Social behavior: what is the rationale for using 2 different arenas to test social behavior (both for imaging and optogenetics studies)? Are the mice tested on the 2 arenas the same or they are from different cohorts? Are the results in the graphs the result of an average between the data from both arenas?

Thanks for the comment. For optogenetic manipulations, we only used the 3-chamber arena, not the round chamber. Figure 7 was corrected showing only the 3-chamber arena for social interactions. The calcium imaging data was acquired in two different chambers as indicated and the data was pooled.

13) Extended Fig. 9 H-I reports food intake, % freezing and distance moved. 9J does not show data points, please uniform this to the other graphs. Further, I would like the author to report the statistical test used and the p-value for each experiment, in every figure of the paper. This information would help the reader to interpret the data.

Thanks for spotting this mistake. We corrected Extended data Fig 9J now showing data points and added p values with all statistics in the legend.

14) Why did the authors opt for a CPP and not for a real-time place aversion or preference (RTPP or RTPA) to test “the intrinsic valence of optogenetic activation”? The authors should test the intrinsic valence of optogenetic inhibition as well. Understanding if inhibition has valence would help the interpretation of optogenetic studies.

Thanks for the comments. We had no particular reason for using CPP instead of RTPP, since both assays measure intrinsic valence of the neurons. We considered it redundant to do both tasks. We agree with the second point and added new data on CPP with optogenetic inhibition from a new cohort of mice in Extended data Fig.12G. In the main text (Line 433) the results were explained: *Conversely, photoinhibition of BLA^{Lypd1} neurons caused a significant avoidance of the photostimulation-paired chamber, whereas photoinhibition of BLA^{Eiv1} neurons resulted in significant preference behavior for the photostimulation-paired chamber (Extended data Fig.12G).*

15) Line 432-433: "...BLA-Lypd1 neurons are positive-valence neurons: they are activated during fasting and food approach behavior...". A negative energy balance (i.e. during fasting) is usually associated with negative valence. Can the authors comment on this in my opinion counter-intuitive conclusion? Are these neurons activated during eating (see point 7)? Food is a positive stimulus, especially to a mouse that is hungry, and would corroborate the positive-valence conclusion.

Thanks for this comment. Although we did not provide direct data for Lypd1 neurons being activated during eating, the fact that the activity of Lypd1 neurons increases when the animal is in the food zone suggests exactly that. Activation of Lypd1 neurons during fasting was observed in tissue slices, where Lypd1 neurons showed higher excitability compared to slices from fed mice. Our interpretation of these data is that palatable food engages Lypd1 neurons. The hungrier the animals are, the more palatable the food becomes. During fasting, Lypd1 neurons are activated and contribute to the process of palatability-guided feeding. Lypd1 neurons therefore have similar roles as recently described central amygdala appetitive neurons (Peters et al 2023; Douglass et al 2017). We added this explanation in the discussion (Line 551):

16) Line 460: "Our snRNAseq dataset did not include Ppp1r1b, which may either indicate low expression levels of this transcript as previously suggested". Zhang et al (Li), Nat. Neuro, 2021 (the reference used to support this conclusion), shows that Ppp1r1b expression is detectable with RNA scope, but these authors do not seem to comment on Ppp1r1b low expression level. Kim et al (Tonegawa), Nat. Neuro, 2016, also shows detectable levels of Ppp1r1b. Therefore, the conclusion that Ppp1r1b levels are not detected in this dataset because of the low level of expression of Ppp1r1b should be revised.

Thanks for bringing this up. We have described the situation of Ppp1r1b more carefully in the discussion (Line 522): *Our characterization of BLA^{Rspo2} neurons corresponds well with previous work³ in terms of their enrichment in the anterior BA and their function in negative-valence behavior. The same study characterized BLA^{Ppp1r1b} as positive-valence neurons being spatially segregated from BLA^{Rspo2} neurons in the posterior BA³. More recent studies challenged this view by showing that the majority of Rspo2+ neurons in the BLAp were Ppp1r1b+ (Zhang et al) and that Ppp1r1b is a marker for GABAergic clusters (Hochgerner study). Our snRNAseq dataset did not include Ppp1r1b, perhaps reflecting limitations of snRNAseq in detecting certain low abundant mRNAs.*

Minor comments

- 1) Line 125: "...and Pde11a 5 (Fig. 1D, E)". Is the name of the gene Pde11a or Pde11a5? Or does it refer to cluster 5?

Thanks for pointing out these mistakes. The name of the gene is Pde11a. We corrected this (Line 127).

- 2) In extended Fig. 9 the text for (H) is missing, and the legend text from there onwards does not correspond to the panels in the figure.

Thanks, we corrected the legend and extended data fig.9.

- 3) Figure 5 C and D: In C, Lypd1 is the top and Etv1 is the bottom panel, in D it is the opposite. I suggest to uniform the order of the panels (e.g. Etv1 as bottom panel in both C and D) and to add the gene names to the panels so that the reader can promptly read the figure.

Well appreciated. We corrected Fig 5 as suggested.

- 4) Figure 5F does not seem to be reported in the main text.

Thanks, we now added 5F in line 361.

- 5) Line 160: "in ppLA (Fig. 2A, D)". I am unclear how to read these figures. I find the choice of using the same colors (i.e., black and yellow) for every pair of genes confusing. I would find it more helpful to assign distinct colors per each gene and to read this figure, rather than from left to right, from top to bottom, so that the reader can appreciate the change in the spatial expression of a given marker along the antero-posterior axis of the BLA. To bring an example of what I got confused, Line 172-173 reads "Otof-positive cells were highly

concentrated at the lateral edge of the alBA and rather scarcely present in the rest of the alBA (Fig.2A)". If I go to the figure, and I interpret it correctly, the first data I see in 2A is relative to *Lypd* and *Etv1*, and not *Otof*.

Previous versions of this figure had distinct colors assigned to each gene, but this did not help with clarity. We therefore changed it back to black and yellow. To improve the readability of the figure, we added more panel numbers for each panel of the pairing of genes such as A1, A2, A3, A4, and A5 for 5 pairs of 10 genes in Fig2. These changes are now used in the main text (from Lines 158 to 190) and in the legend. Therefore, now the reader can be guided more easily to Fig.2 A4 to see *Otof*-positive cells (line 176): *Otof*-positive cells were highly concentrated at the lateral edge of the alBA and rather scarcely present in the rest of the alBA (Fig. 2 A4): Below is Fig. 2A panel and *Otof* expressing cells are shown in A4 panel

- 6) Line 256: "Basic firing rates did not differ between cells (Figure S9A)", but Figure S9A shows a significant difference. Can the author comment on this?

We thank the reviewer for pointing this out. While it is true that there is a significant difference in the firing rate between *Rspo2* and *Lypd1* cells in one single current step, it is not for the other steps and thus we mentioned that they do not differ. To avoid any misunderstandings, we now rephrased the sentence: "Basic firing rates did not differ greatly between cells (Figure S9A)" in Line 262.

Reviewer #2:

Comments for the Author:

Understanding the molecular and functional diversity among neuronal subgroups in the basolateral amygdala (BLA) can provide important insights into how this brain area encodes positive and negative valence and regulates valence-related behaviors. Some recent studies have performed single-cell RNA sequencing on the BLA (O'Leary 2020; Hochgerner 2023). In this study, Lim et al. performed single-nucleus RNA sequencing (snRNA-seq) to identify molecularly defined neuronal subgroups in BLA and conducted in situ hybridization of 10 selected genes to map the spatial distribution of glutamatergic clusters across different subregions of the BLA. They then performed detailed functional characterization of three glutamatergic subpopulations (*Lypd1*+, *Etv1*+, and *Rspo2*+) using electrophysiological recording, microendoscopic imaging, and optogenetic manipulation. Their results suggest that these subpopulations respond differentially to positive/negative valence and social cues and play distinct roles in regulating food uptake, fear memory, and social interaction. The data and findings of this study potentially represent a valuable resource for the field. However, several major issues need to be addressed.

Major points:

1. The number of cells sequenced in this study is relatively small (~4500 neurons) compared to other recent single-cell RNAseq studies in the amygdala (e.g. ~30,000 neurons in Hochgerner 2023, Nat. Neurosci.). In addition, only cells from male animals were included. These constrain the power and resolution of the analysis and diminish the value of this dataset as a resource for the field. Considering the relatively large size and heterogeneity of the BLA and the relative ease of sequencing a large number of cells using the 10x Genomics platform, augmenting the study by sequencing more cells from both male and female animals could significantly enhance its importance.

It is true that the Hochgerner study included 30,000 neurons, but their aim was to characterize all amygdala nuclei (our study focused on BLA only) and from mice that had undergone fear conditioning plus the relevant control mice (our data is from normal mice only). In fact, after extracting the data from excitatory BLA neurons from the Hochgerner study, we found that our study included more excitatory BLA neurons than the Hochgerner study (n=1960 versus n=1185, see table below) and our dataset contained a larger number of unique cells (n=697 versus n=82).

	VGLUT1-1	VGLUT1-1	VGLUT1-1	VGLUT1-2	VGLUT1-2	VGLUT1-2	VGLUT1-3	VGLUT1-3	VGLUT1-6	VGLUT2-2	Cells unmatched to HG clusters in our data	Cells matched to HG clusters in our data			
Etv1 (am/alBA)	0	1	0	2	37	0	0	0	163	0	45	248			
Etv1/Lypd1 (acBA)	0	0	63	1	0	1	0	0	3	0	10	78			
Grik1 (acBA,pLA)	0	0	0	0	3	0	0	0	5	0	41	49			
Otof (pBA)	8	42	4	6	0	2	66	1	19	0	106	254			
Otof/Lypd1/Cdh13 (pLA,pBA)	1	5	55	2	13	50	1	0	9	0	105	241			
Rorb (aLA)	0	4	0	4	7	2	3	0	8	0	119	147			
Rorb/Lypd1 (aLA,pLA)	0	0	0	0	129	0	0	0	0	0	12	141			
Rspo2 (am/alBA)	0	0	0	1	0	0	0	0	11	0	57	69			
Rspo2/Etv1/Adams2/Bdnf (aB	0	2	0	196	4	1	7	0	5	0	52	267			
Sema5a (pLA)	0	0	0	0	1	4	0	0	1	71	131	208			
Sema5a/Otof/Lypd1/Cdh13 (p	0	0	0	0	110	126	3	0	0	0	19	258			
											697	1960	all cell numbers in our BLA glutamatergic clusters.		

	Etv1 (am/	Etv1/Lypd	Otof (pBA	Otof/Lypd	Rorb/Lypc	Rspo2/Etv	Sema5a (p	Sema5a/C	Cells unmatched to our clusters inHG	Cells matched to our clusters in Hochgerner data			
VGLUT1-13-Dcn_C1ql2	0	40	12	5	0	0	0	0	44	101			
VGLUT1-14-Cartpt_Fam46a	0	5	5	1	0	0	0	0	5	16			
VGLUT1-15-Fmo1_Rxfp3	0	70	3	15	0	0	0	4	12	104			
VGLUT1-2-Rspo2_Sema3e	6	1	1	0	0	362	0	0	1	371			
VGLUT1-28-Grp_Cpne8	17	1	0	1	150	3	0	86	4	262			
VGLUT1-29-St8sia2_Cald1	0	0	0	2	0	0	0	19	1	22			
VGLUT1-3-Sema5a_Dcn	0	1	25	0	1	38	0	10	4	79			
VGLUT1-32-Mid1_Cdh22	1	0	2	9	0	0	0	7	2	21			
VGLUT1-6-Thrsp_Lamp5	180	2	0	11	0	1	0	3	3	200			
VGLUT2-24-Cdkn1c_Penk	0	0	0	0	0	0	3	0	6	9			
									82	1185	all cell number of HG BLA cluster		

As requested by reviewers 1 and 3, we have included in the revised manuscript a detailed comparison of our data with the Hochgerner data in the discussion (line 507) (Extended data Figure 14, below). We agree with the reviewer that future transcriptomics studies should include a comparison of cells from male and female animals. Adding cells from female animals to our study including their spatial annotation with smFISH would, however, take a lot of time and, considering the competitive situation in the field, would not be advisable.

2. It was mentioned In the Methods section that both male and female animals were used for the smFISH analysis and functional characterizations. However, the data from both sexes seem to be merged in all figures, with no indication of whether the results were similar or different between males and females. It would be beneficial to include a comparison between the two sexes. At a minimum, plotting the results from males and females separately could provide a clearer picture.

We agree and plotted the results from males and females separately in Extended data fig.12 A-C. We found there is no difference in sex, although group sizes became rather small (Line 467) like below:

Extended Data Fig.12 related to figures 6-7: no sex difference

(A) Food intake during optogenetic manipulation of BLA^{Lypd1} and BLA^{Etv1} populations compared to light-off epochs in male and female mice. ChR2 groups (top): Lypd1 group, Male: n=8, Female: n=6 with $F(3, 24) = 2.301$, $P=0.1028$, Etv1 group: Male: n=6, Female: n=5 with $F(3, 18) = 2.201$, $P=0.1231$; Tukey's multiple comparisons test, Bonferroni post hoc analysis. eNpHR 3.0 groups (bottom): Lypd1 group, Male: n=7, Female: n=3 with $F(3, 34) = 1.879$, $P=0.0598$, Etv1 group: Male: n=4, Female: n=4 with $F(3, 12) = 0.7522$, $P=0.7624$; Tukey's multiple comparisons test.

(B) Cumulative duration in the social zone (%) during optogenetic manipulation of BLA^{Lypd1} and BLA^{Etv1} populations compared to light-off epochs in male and female mice. ChR2 groups (top): Lypd1 group: Male: n=4, Female: n=4 with $F(3, 12) = 1.647$, $P=0.2309$; Etv1 group: Male: n=6, Female: n=6 with $F(3, 20) = 0.9957$, $P=0.4151$ Tukey's multiple comparisons test, Bonferroni post hoc analysis. eNpHR 3.0 groups (bottom): Lypd1 group, Male: n=5, Female: n=2 with $F(3, 10) = 0.5327$, $P=0.0598$, Etv1 group: Male: n=8, Female: n=4 with $F(3, 18) = 0.2474$, $P=0.3694$; Tukey's multiple comparisons test.

(C) Freezing behavior (%) on day 2 combined with photo-manipulation of BLA^{Lypd1} and BLA^{Etv1} populations compared to males and females. ChR2 groups (top Lypd1 group: Male: n=4, Female: n=3 with $P=0.9989$, Etv1 group: Male: n=4, Female: n=3 with $P=0.6284$; eNpHR 3.0 groups (bottom): Lypd1 group, Male: n=3, Female: n=2 with $P=0.1666$, Etv1 group: Male: n=3, Female: n=3 with $P=0.9327$; Unpaired t-test.

- Fig. 2: When matching smFISH cells to the transcriptomic clusters, each cell was assigned to one of the 11 clusters according to the highest correlation coefficient. It's unclear whether any threshold was set for the correlation coefficient and significance level. Were there any cells that could not be reliably assigned to any cluster?

To avoid any bias, we did not set any threshold and forced all cells to be assigned to one of the clusters.

- Fig. 2F: Please clarify how the grouping of subregions was performed. It appears that the correlation between LAa and LAp is higher than that between LAp and LApp, but LAp was grouped with LApp instead of LAa. Similarly, correlation between BAal and BAac is higher than that between BAac and BAp, but BAac was grouped with BAp instead of BAal.

Thanks for the comment. We agree. The grouping of subregions (red boxes in 2F) creates confusion and is not very helpful. Since it is a minor point, we removed the red boxes from Fig. 2F.

- Fig. 6C-E, 7C, D: Two-way ANOVA with post hoc tests should be used.

We thank the reviewer for this suggestion. We used paired t-test to compare the same mice across light on/off days as done in other studies (Kim et al 2016). We added statistics using one-way ANOVA from all optogenetic experiments from three different cre lines in Table 1. The one-way ANOVA also showed consistency with our t-test results. We also added two-way mixed-effects ANOVA followed by Tukey's multiple comparisons test in Table 1. The main results were consistent across three different statistical methods.

For example below:

Food-chr2						CPP-chr2						CPP-Nphr					
Tukey's multiple c	Mean Diff.	95.00% C	Below thr	Summary	Adjusted P Value	Tukey's m	Mean Diff.	95.00% C	Below thr	Summary	Adjusted P Value	Tukey's m	Mean Diff.	95.00% C	Below thr	Summary	Adjusted P Value
on:Chr2 vs. off:Y	0.1364	0.02364 to	Yes	*	0.0174	pre:Chr2	-14.2	-42.27 to	No	ns	0.4013	pre:eNpHf	19.42	-18.74 to	No	ns	0.397
on:Chr2 vs. off:C	0.08798	0.000698 to	Yes	*	0.0477	pre:Chr2	-32.24	-53.37 to	Yes	**	0.0033	pre:eNpHf	60.07	14.78 to	Yes	*	0.0103
on:Chr2 vs. off:Y	0.1008	-0.02206 to	No	ns	0.1208	pre:Chr2	-19.47	-52.70 to	No	ns	0.2943	pre:eNpHf	37.01	-2.857 to	No	ns	0.0681
on:YFP vs. off:Ch	-0.04642	-0.09813 to	No	ns	0.0569	pre:YFP v	-18.03	-41.29 to	No	ns	0.1323	pre:mCher	40.65	8.247 to	Yes	*	0.0174
on:YFP vs. off:YF	-0.03577	-0.08612 to	No	ns	0.2007	pre:YFP v	-5.269	-17.70 to	No	ns	0.5564	pre:mCher	17.59	-3.377 to	No	ns	0.1062
off:Chr2 vs. off:Y	0.01265	-0.04430 to	No	ns	0.9067	post:Chr2	12.76	-15.58 to	No	ns	0.4899	post:eNpH	-23.06	-64.50 to	No	ns	0.3308

Food-nphr						social-chr2					
Tukey's multiple c	Mean Diff.	95.00% C	Below thr	Summary	Adjusted P Value	Tukey's m	Mean Diff.	95.00% C	Below thr	Summary	Adjusted P Value
on:eNpHR 3.0 vs.	-0.00707	-0.05992 to	No	ns	0.9778	on:Chr2 v	-12.88	-42.79 to	No	ns	0.4595
on:eNpHR 3.0 vs.	-0.05976	-0.1126 to	Yes	*	0.0234	on:Chr2 v	-16.82	-36.65 to	No	ns	0.0992
on:eNpHR 3.0 vs.	-0.01407	-0.08155 to	No	ns	0.924	on:Chr2 v	-5.994	-24.32 to	No	ns	0.6488
on:mCherry vs. of	-0.05269	-0.1109 to	No	ns	0.081	on:YFP vs	-3.942	-41.55 to	No	ns	0.9783
on:mCherry vs. of	-0.007	-0.04396 to	No	ns	0.9474	on:YFP vs	6.886	-27.22 to	No	ns	0.8753
off:eNpHR 3.0 vs.	0.04569	-0.02159 to	No	ns	0.2355	off:Chr2	10.83	-13.89 to	No	ns	0.4472

social nphr					
Tukey's m	Mean Diff.	95.00% C	Below thr	Summary	Adjusted P Value
on:eNpHR	21.47	-9.241 to	No	ns	0.1434
on:eNpHR	6.205	-12.61 to	No	ns	0.6802
on:eNpHR	11.8	-20.61 to	No	ns	0.5212
on:mCher	-15.27	-42.18 to	No	ns	0.2389
on:mCher	-9.672	-36.03 to	No	ns	0.516
off:eNpHR	5.594	-23.43 to	No	ns	0.8583

6. Fig. 7B: There was a notable trend of increased freezing following inhibition of Lypd+ neurons, but the result did not reach statistical significance due to the small sample size (5 animals). This result should be confirmed with a larger sample size.

We agreed and performed one more experiment with a new cohort of mice to increase the sample size in Fig 7B. We found that there was indeed an increase in freezing (%) in the Lypd1 neuron-inhibited group compared to the control (Line 449) like below:

Minor points:

1. Fig. S2C: Please clarify how cell boundaries were determined in smFISH, given that only a nuclear marker but no membrane marker was used.

Thanks for the comment. Based on nuclear detection, putative cell boundaries were created by the HALO algorithm with a given parameter (8 μm around the nucleus to target 12 μm of cell diameter). This follows the assumption that an average BLA glutamatergic neuron has a diameter of 10-13 μm (Kim et al 2016, 2017). The explanation was added to the legend of Extended data Fig. 2C.

2. Fig. 4-5: The number of mice used for the imaging experiments should be indicated, in addition to the number of cells.

The numbers of mice used for each imaging experiment are now indicated in the legends of Fig. 4 and 5.

3. Fig. 4B: Please specify what the green and gray traces each represent.

Thanks for the comment. During current clamp recording (step-wise current injections), gray represents all the current steps applied to these neurons, while the green trace is one recording highlighted to see the difference in the firing response in fed and fasted animals after the injection of 230 pA current step. We added this detailed information to the legend of Fig.4B

4. Fig. 7D: Data for the mCherry control group for Rspo2+ neurons were missing.

This control group was left out when we noticed that all other groups of Rspo2+ neurons did not show any differences in the social interaction assay.

Reviewer #3:

Comments for the Author:

In their manuscript, Lim et al. identified candidate marker genes for several clusters of glutamatergic neurons in the BLA by cross referencing RNA sequencing data and ISH data from the Allen atlas, then outlined the anatomical expression patterns of these genes and the neurons they demarcate in situ via mFISH. They then used genetically targeted in vivo calcium recordings and ex vivo electrophysiology to observe differences in neuron activity upon food approach and social interaction for a subset of glutamatergic neurons. The authors then leveraged optogenetic manipulation of these neurons to show that glutamatergic BLA neuron subtypes help mediate feeding behavior, social interaction, and fear memory formation.

Overall, the experiments presented in this manuscript are challenging, and the authors should be commended for the sheer volume of work here. The manuscript follows a logical order through the experiments from transcriptomics, to slice properties, to in vivo imaging and manipulation. However, as written, the manuscript struggles to differentiate itself from other recent work in the field including O'Leary et al 2020 and Hochgerner et al 2023, which brings into question the novelty of some of the results. In terms of the novel functional work, the interpretation of the data presented here requires further consideration, as several results, though interesting, are overstated and require further justification. Relative to existing work, the novel data presented here will be of great interest to a select audience, but the general interest of this work may be limited.

Major Comments:

1. This manuscript reads largely like two disjoint manuscripts. The first three figures are devoted to transcriptomics, whereas the remaining figures are devoted to behavior and are not really rooted in any transcriptomics work beyond exploiting markers. I'll discuss these two elements (transcriptomics and behavior) separately.

Transcriptomics: A lot of the first three figures seems to be relatively incremental compared to previous work - specifically (O'Leary et al eLife 2020) and (Hochgerner et al Nat Neuro, 2023). The former is cited and directly compared; the latter is not, but this is likely attributable to Hochgerner et al publishing their paper right as Lim et al submitted their work. Relative to these publications, there is a significant amount of overlap in terms of discovering the "baseline" cell types in the basolateral amygdala. I do not feel that the transcriptomics work here is a significant advancement relative to previous publications - it is deeper and more comprehensive than O'Leary, but to my reading offers similar findings and no new general principles relative to Hochgerner. At minimum, given the recent Hochgerner publication, the authors

here must do a careful direct analysis relative to Hochgener and demonstrate emergent new principles relative to this published work for there to be a significant advance.

We thank the reviewer for these insightful comments. It is true that our study is deeper than the O'Leary study. In fact, our study is also deeper than the Hochgener study. After extracting the data from excitatory BLA neurons from the Hochgener study, we found that our study included more excitatory BLA neurons than the Hochgener study (n=1960 versus n=1185, see table below) and our dataset contained a larger number of unique cells (n=697 versus n=82).

	VGLUT1-1	VGLUT1-1	VGLUT1-1	VGLUT1-2	VGLUT1-2	VGLUT1-3	VGLUT1-3	VGLUT1-6	VGLUT2-2	Cells unmatched to HG clusters in our data	Cells matched to HG clusters in our data	
Etv1 (am/alBA)	0	1	0	2	37	0	0	0	163	0	45	248
Etv1/Lypd1 (acBA)	0	0	63	1	0	1	0	0	3	0	10	78
Grik1 (acBA,pLA)	0	0	0	0	3	0	0	0	5	0	41	49
Otof (pBA)	8	42	4	6	0	2	66	1	19	0	106	254
Otof/Lypd1/Cdh13 (pLA,pBA)	1	5	55	2	13	50	1	0	9	0	105	241
Rorb (alA)	0	4	0	4	7	2	3	0	8	0	119	147
Rorb/Lypd1 (alA,pLA)	0	0	0	0	129	0	0	0	0	0	12	141
Rspo2 (am/alBA)	0	0	0	1	0	0	0	0	11	0	57	69
Rspo2/Etv1/Adamts2/Bdnf (aB.	0	2	0	196	4	1	7	0	5	0	52	267
Sema5a (pLA)	0	0	0	0	1	4	0	0	1	71	131	208
Sema5a/Otof/Lypd1/Cdh13 (pp	0	0	0	0	110	126	3	0	0	0	19	258
											697	1960 all cell numbers in our BLA glutamatergic clusters.

	Etv1 (am/	Etv1/Lypd	Otof (pBA	Otof/Lypd	Rorb/Lypc	Rspo2/Etv	Sema5a (p	Sema5a/C	Cells unmatched to our clusters inHG	Cells matched to our clusters in Hochgener data
VGLUT1-13-Dcn_C1ql2	0	40	12	5	0	0	0	0	44	101
VGLUT1-14-Cartpt_Fam46a	0	5	5	1	0	0	0	0	5	16
VGLUT1-15-Fmo1_Rxfp3	0	70	3	15	0	0	0	4	12	104
VGLUT1-2-Rspo2_Sema3e	6	1	1	0	0	362	0	0	1	371
VGLUT1-28-Grp_Cpne8	17	1	0	1	150	3	0	86	4	262
VGLUT1-29-St8sia2_Cald1	0	0	0	2	0	0	0	19	1	22
VGLUT1-3-Sema5a_Dcn	0	1	25	0	1	38	0	10	4	79
VGLUT1-32-Mid1_Cdh22	1	0	2	9	0	0	0	7	2	21
VGLUT1-6-Thrsp_Lamp5	180	2	0	11	0	1	0	3	3	200
VGLUT2-24-Cdkn1c_Penk	0	0	0	0	0	0	3	0	6	9
									82	1185 all cell number of HG BLA cluster

After integrating the data from the Hochgener study with our data, we found that several of the main clusters matched well with our clusters (Extended data Fig. 14A, B). For example, our BA cluster 1 (Rspo2/Etv1/Adamts2/Bdnf) showed a near-perfect match with Hochgener's VGLUT1-2 cluster from anterior BA (Extended data Fig. 14C, D). Our clusters from the LA (cluster 2, Sema5a/Otof/Lypd1/Cdh13 and cluster 8, Rorb/Lypd1) showed similarity to Hochgener's VGLUT1-28 and VGLUT1-29 clusters from LA (Extended data Fig. 14C, E). A comparison of genes matching in both datasets revealed that the pattern of gene expression was distinct between the BA and LA clusters supporting the idea that these subregions contain genetically-defined subpopulations (Extended data Fig. 14D, E). Some of our most anterior and most posterior located clusters (our clusters 5 and 7) remained unassigned in the Hochgener subset. Since part of the spatial distribution in the Hochgener study was inferred by correlation with the Allen Mouse Brain Atlas, we speculate that these clusters were wrongly assigned to non-BLA amygdala clusters. We have added this analysis to the discussion (line 507) like below:

Additionally, it would be important if the authors could somehow use the genes expressed within these cell types to better understand cell type function or make cell-type perturbations examined in behavior. Right now, the transcriptomics work is really just used to identify marker genes for distinct amygdalar neurons, with several of these markers known already from previous work. If the authors could more closely integrate their transcriptomics and behavior work this would significantly strengthen this manuscript.

Thanks for the comment. We tried to find unique genes that would mark individual glutamatergic cell clusters for subsequent spatial and functional analysis. Unfortunately, glutamatergic unlike GABAergic cell clusters share many of their top marker genes, and we could only identify distinctive marker combinations. These markers were then used for spatial annotations. For functional analysis, we used three Cre lines each targeting between two and three spatially annotated cell clusters. Two of the Cre lines are novel and the third Cre line serves as a reference. We would respectfully argue that the combination of transcriptomics work, spatial annotation, and subsequent functional analysis including calcium imaging and behavior, is a package of results that is appealing not only to specialists but also to the interested reader with any background in neuroscience.

Behavior. The behavior in this manuscript is extensive, which I laud the authors for. However, in many ways the foundational aspects of the behavior datasets need additional info, specifically regarding the properties of the mouse lines used, which do not seem to be well-characterized. The authors examine a reporter cross that is considered relative to the endogenous gene (Fig. S8), but to my eyes the labeled cells look very different between the two cases. I cannot see how this would result in the 80%+ percentage colabeling the authors report - for example, the *Etv1* endogenous labeling looks to be at least twice as much as the transgenic cross reporting. Additionally, there is no imaging or quantification of viral reporting (which is the primary approach for behavior), and which may differ significantly from reporter crosses.

Thanks very much for the comment. We have confirmed that the quantification was correct and have reformatted this figure with more representative images as well as high-magnification examples. The expression of *Etv1-Cre* is less than endogenous *Etv1* because the Cre line is Tamoxifen inducible. An example of how we quantified mFISH and Tom expression was added to Extended data Fig.8 E. White arrows indicate cells positive for both Tom and endogenous mRNA signal, empty arrows mark cells positive only for endogenous mRNA signal, asterisks indicate cells that are not considered for quantification due to lack of a nuclear signal (Dapi).

In our hands, viral reporting usually follows well the spatial pattern of genetic reporters but is more variable in terms of the number of cells labeled. We therefore felt that quantification of genetic reporter expression would suffice.

This should also be done, and compared to both the transgenic cross as well as the endogenous genes. This is particularly important as both *Etv1* and *Lypd1* seem to be expressed in inhibitory cells (Hochgener et al., and <https://zeisellab.org/amygdala/>), and thus imaging and optogenetic manipulation may be influenced by off-target cell types. The authors should clearly demonstrate that their approaches are restricted to the excitatory populations they seek to image/manipulate, otherwise this body of work becomes challenging to interpret.

Our snRNAseq data showed that *Etv1*, *Lypd1*, and *Rspo2* are mainly localized in glutamatergic, rather than GABAergic clusters. We have included violin plots of *Etv1* and *Lypd1* positive cells in glutamatergic and GABAergic neurons in our snRNAseq data to support this claim (*Lypd1* portion: 71% in Glu and 29% in GABA cells). We have also included a quantification of *Lypd1-Cre*;tdTom expression with *Slc17a7* (*Vglut1*) showing more than 80% co-labeling like below:
 Line 255: Additionally, our snRNAseq data showed that *BLA^{Etv1}* and *BLA^{Lypd1}* neurons are mainly localized in glutamatergic, rather than GABAergic clusters (Extended Data Fig. 8 F) and quantification of *Lypd1-Cre* or *Etv1-CreER*; tdTomato expression with *Slc17a7* (*VGLUT1*) showed more than 80% co-labeling (Extended Data Fig. 8 G, H).

2. The writing and presentation in this manuscript is generally unclear. I'll list several examples below, but broadly the writing and figures should be made more streamlined and linear.

- The introduction moves back-and-forth between background and manuscript-investigated lines of inquiry, and it is hard to get "oriented" to the goals of the paper from this approach

- Legend panels are occasionally listed out of order (example: "E" before "D" in Fig 4)

- Figure panels should have labels to clarify what they are showing (example: Fig 5C,J,K,L should all have genotype labels)

- Several statements seem to contradict their preceding topic sentences ("The results matched with previously published work" (line 405) precedes the sentences "cluster 1... which localized to the whole anterior-posterior extent of the BA, was included in all O'Leary et al.'s BA1-BA4" (line 449))

We are grateful to the reviewer to for pointing out these deficiencies.

1. We reorganized the introduction and added a paragraph on the Hochgerner study (Line: 88)

2. Legend of Fig.4: the order of panels was corrected.

3. We added genotype labels to figure panels.

4. We toned down the statement: "The results matched pretty well with previously published work".

7. The authors seem to be making arguments that some neural cell types, but not others, are important for various types of behavior. However, it does not appear that there are direct statistical comparisons between genotypes; rather, all statistical comparisons appear to be within-genotype. The authors should to do across-genotype statistical comparisons in order to implicate specific cell types in their assayed behaviors.

This is a very good point, thanks. We did perform across-genotype comparisons with the calcium imaging data using a permutation test that mixed all neurons regardless of genotype. For the behavior, we compared the three mouse lines only to their respective control groups (as is usually done).

We now added comparisons across genotypes for optogenetic manipulation data in Extended data Fig12 D-F. In Line 469, we explained the results: *When comparing the results across genotypes, we found that social interactions and freezing behavior were significantly promoted by BLA^{Etv1} neuron activation in comparison to BLA^{Lypd1} neuron activation, and similar effects were observed by BLA^{Lypd1} neuron inhibition compared to BLA^{Etv1} neuron inhibition. The opposite effects across genotypes were seen for the promotion of food consumption (Extended data Fig. 12D-F).*

Minor comments:

1. Fig 1C: doesn't show actual expression - only phenotyping. The authors should show overall expression of excitatory and inhibitory genes.

Thanks for the comment. We added the expression plots in Extended data Fig. 1C

2. Figure 5B: I do not understand how trial number can be in the hundreds, please clarify. What do black and grey here denote? Please supply a range bar if this is illustrating a continuous variable.

Thanks for the question. Footshocks were given three times per session. Trial numbers are neuron numbers times three footshocks. The spikes for the PSTH plot were inferred from calcium traces. The color indicates the normalized spike rate (black=max, white=min rate). The range bar was added in Fig. 5B. The trial number was explained in the legend of Fig.5B (Line 1446)

1. Figure 4C: Lypd1 differences in excitability in fasted vs. fed - can this simply be explained by different in resting membrane potential? Leftward shift in fasting suggests may be the case. The authors should correct for resting membrane potential and re-run their analysis.

Yes, we found the resting membrane potential altered after fasting which means the electrophysiological properties changed only in Lypd1 neurons. In brief, an increase in the firing rate of the fasted Lypd1 neuron occurs because of the depolarization of the membrane potential. This brings the neuron closer to the threshold for firing. Hence, we conclude that Lypd1 neurons are more excitable after fasting. We will rephrase this part to make it clearer. A better explanation was added in the main text (Line 270):

Since the depolarization of the membrane potential brings the neuron closer to the threshold for firing, this contributes to an increase in the firing rate of the fasted BLA^{Lypd1} neuron. This suggests that BLA^{Lypd1} neurons are more excitable after fasting.

4. Shock response scores are abbreviated "SRC", maybe better fit is the direct acronym "SRS"?

Thanks for the suggestion, but we prefer to keep SRC.

5, Fig. 5D: The authors make claims based on there being an effect in these data; however, neither fit meets significance, and thus one cannot claim to have any effect.

We agree and have rephrased the sentence in the main text line (Line 346): *we found a positive although not statistically significant trend in the BLA^{Etv1} population ($R^2 = 0.6$), but no such trend in the BLA^{Lypd1} population (Fig. 5D).*

6. Fig. S9I: freezing looks the same in these group data, but many more mice here than in main figure (Fig. 5D). Additionally, the mice in Fig 5D also have very different freezing levels, why is this the case? Can the authors be confident in that effects measured reflect the cell types rather than animal-to-animal variability in the main figure?

Thanks for the comment. We have explained this better in the methods (line 1005): *we only included mice having neuronal data on both Day 1 and 2. One (Etv1-Cre) and two (Lypd1-Cre) mice did not have good quality-neuronal data either on Day1 or Day 2 after preprocessing. These mice were excluded from neuronal data analysis.*

7. Small typo of "Chr2" rather than "ChR2" (Fig 6c)

Thanks, noted.

8. New anatomical abbreviations are introduced in the following line: "to divide the BLA into eight subregions from anterior to posterior (aLA, pLA, ppLA, amBA, alBA, acBA, pBA, ppBA) according to published methods 51(Fig. 2A-D)". However, the citation provided does not contain descriptions of several of the abbreviations, and the abbreviations should be included in the present manuscript regardless.

Thanks for pointing out this omission. All subregions except ppLA, ppBA were used in the citation. The citation did not mention the terms ppLA and ppBA, but they analyzed these very posterior parts distinct from other parts of the BLA. Therefore, we named these very posterior parts as 'pp'. We prepared Table 3 with the abbreviation and we added this explanation in Method (Line 766).

9. I would remove "innate" from the sentence "We characterized a positive-valence subpopulation that increased activity in response to the presence of food and promoted innate feeding" as it is unnecessary, since feeding is, by definition, an innate behavior.

Thanks for the point. We rephrased in Line 99 as suggested.

10. Figure 4 P-R: the text formatting makes the numbers "0" and "50" on the y-axis squished.

Thanks, we corrected them.

REVIEWER COMMENTS

Reviewer #1 (Remarks to the Author):

The authors discussed extensively all the points I have raised, and I am generally satisfied with their response.

I have just some minor comments:

- Answer to question 4c reads “Lypd1-expressing cells showed little overlap with Etv1-...”. Further in question 4f, the author's comment reads “...the extent of colocalization...”. Are the terms “overlap”, “colocalization” and “occupancy” (Ext. fig 7) all used to describe the anatomical position of the cells rather than the fact that distinct molecular markers are expressed in the same cell (i.e. co-expression)? If so, and to avoid misunderstandings, I would opt for the use of “colocalization (within an area)”, rather than “overlap”, when referring to the anatomical position of a cell.

We agree with your point. We rephrased “overlap” to “colocalization in BLA”. (Line 244 to 248)

- Point 7 and 8. One of the main points I have raised is relative to the fact that the authors used the approach to food as a proxy for eating (rather than just measuring eating). However, the approach might or might not result in food ingestion. The authors now show that in their paradigm, when the mouse approaches the food area (within 5cm) is very likely to ingest food (point 7). This explanation is reasonable to me. However, if the authors want to use the term “feeding” rather than “approach to food” it should be reported in the main text that “feeding”, in their paradigm, is not just the act of eating (ingesting) but rather includes multiple phases: 1) the approach to the food (as a main parameter), 2) subsequent lifting/handling of the pellet, and 3) ingesting the pellet (point 8).

Thanks a lot for clarifying this point. As suggested, we added in the main text: “The feeding assay was the same as the one used for calcium imaging and included approach to the food, subsequent lifting/handling and ingestion of the pellet.” (lines 414-415).

- Point 12. I feel it is still not clear why using 2 different arenas (Fig. 5 and Fig. 7) to measure the same behavior (socialization).

We first started the optogenetic experiments with the social 3 chamber assay to also measure the duration in the non-social zone as a control to be compared with the duration in the social zone (Extended Data Fig. 12E, below)

Here, we confirmed that non-social response behavior did not change during photoactivation.

For calcium imaging, a high temporal/ spatial resolution was required to synchronize behavioral and calcium tracing data. Therefore, we found a solution in the recent paper “Simultaneous analysis of social behaviors and neural responses in mice using a round social arena system” (Kim et al., 2022) using a round social chamber with a 180-degree fish eye camera above the wired social cup. It allowed us to capture and analyze social interactions more precisely with minimal blind spots. We have added this information to the methods section (lines 893-894 and 902-903)

- Point 14. The authors added, as I asked, experiments relative to the valence of the photoinhibition of Lypd1 and Etv1 neurons. These new data strengthen their conclusions regarding the involvement of Lypd1 in the regulation of feeding (inhibition of Lypd1 has negative valence and in line with this, when tested for feeding,

the mice eat less). The data about photoactivation/photoinhibition of Etv1 are less intuitive. Photoactivation of Etv1 has a negative valence, while photoinhibition induces place preference. However, photoactivation promotes socialization (generally rewarding), while photoinhibition reduces socialization. These results should be interpreted keeping in mind that Etv1 contains more than one population (Ext. Fig. 10, one responding to shock, one to socialization, one to both) and that optogenetic manipulation will affect the activity of all cells together. The authors should discuss these aspects (the valence of the manipulations in CPP tests and how they relate to behavioral changes in feeding, freezing, and social behavior) more in detail.

We agree with this reviewer on the more complex situation with BLA^{Etv1} neurons and have therefore revised the summary of BLA^{Etv1} neuron functions in the discussion: "In line with the calcium imaging data, our optogenetic manipulation experiments revealed that BLA^{Etv1} neurons are mainly negative valence neurons promoting defensive behavior (contextual fear memory formation and conditioned place aversion) and reducing food intake, in line with previous reports on BLA functions^{35,36,66-68}. However, BLA^{Etv1} neurons also promote social behavior which we interpret as being rewarding, a surprising observation, considering that most of the previous evidence suggested that the BLA mediates aversive aspects of social interaction^{19,23}." (lines 583-589)

We hope that this will clarify things.

We have also added the following sentence as suggested by the reviewer: "Our calcium imaging data indicates that BLA^{Etv1} neurons consist of more than one population, one responding to shock, one to social cues, and one – the largest one – to both. Part of the mixed selectivity of these neurons may arise from the fact that optogenetic manipulation will affect all populations together. Further work is therefore needed to functionally characterize the BLA^{Etv1} subpopulations and their associated microcircuits which are likely very complex." (Lines 597-601).

Reviewer #3 (Remarks to the Author):

The authors have satisfactorily addressed the majority of my comments with the revised version of the manuscript, but there remains one concern outlined below that must be addressed before the associated manuscript should be considered for publication. There is a key set of characterization experiments that are needed to ensure that the transgenic lines are appropriate for the experiments and interpretations; without this, the manuscript runs the risk of misinterpreting manipulated cells and the sources of behavioural phenotypes.

Major Comment:

The authors' response did not address my point regarding an admitted lack of characterization of the viral model: "In our hands, viral reporting usually follows well the spatial pattern of genetic reporters but is more variable in terms of the number of cells labeled. We therefore felt that quantification of genetic reporter expression would suffice." Without this characterization, this makes interpreting the results of the optogenetics experiments difficult. Firstly, reporter crosses can vary substantially from viral reporting due to nature of reporting (crosses permanently report expression of marker gene at any time; viruses report expression roughly only at the time of injection and are shaped by the type and properties of the virus). With no data that validates that Lypd1, Etv1 or Rspo2 cells were successfully targeted for optogenetic manipulation, and that these cells are restricted to the BLA, we cannot be sure that these are the cell types that are responsible for the effects that the authors attribute to them. High enough viral titre can always induce off-target expression of opsins, random insertions (like those used in the Lypd1-cre line from GENSAT BACs) can drive expression in off target cells, and viral spillover outside of the BLA can also contribute to off-target effects. The cell-type specificity, both to the "ground truth" marker gene of interest as well as restriction of expression to the BLA, must be properly quantified to verify that each of the AAV-mediated opsin expressions are faithful to the cell types that are suggested to have been targeted for optogenetic manipulation.

We have addressed the reviewer's concerns by adding the requested data for Lypd1-Cre and Etv1-CreER lines (Extended data Fig. 9). Adult Etv1-CreER and Lypd1-Cre mice were injected with a Cre-dependent virus (AAV5-hSyn-DIO-mCherry) into the BLA and the expression of mCherry fluorescence was compared to endogenous Etv1 and Lypd1 mRNA expression by FISH, respectively. Overall, we observed very little ectopic expression of

mCherry in cells not expressing the endogenous marker. On average, 80% of Etv1-CreER::mCherry-positive cells also expressed Etv1 mRNA. Similar results were obtained for Lypd1 (75%). Conversely, the numbers of Etv1-CreER::mCherry and endogenous Etv1 double positive cells were overall smaller than the Etv1 population (65%). Lower numbers were obtained for Lypd1 (45%). These numbers indicate that the Cre lines target the correct neuron subpopulations. The numbers are somewhat lower than the numbers obtained with a genetic reporter. We assume that this is mostly due to inefficient viral transduction.

Regarding viral spillover outside the BLA, we have optimized all possible factors (virus titer, volume and coordinates of injection) to avoid off-target effects.

In main Figure 6B (see above), we have shown representative images of the typical viral expression pattern. In the 'reporting summary', we specified that any data from mice exhibiting off-target expression outside of the BLA or showing weak expression within the BLA were excluded from our datasets. This has been analyzed by two authors to avoid human bias. We have now included histology images from vibratome slices in Extended data Fig. 9E (below) to provide examples of viral expression confined to the BLA.

Minor Comment:

Regarding the request for clarification about how hundreds of trials were achieved in Figure 5B, the authors did not clarify how many animals and how many individual trials were performed, which could have been provided in a single line: x animals * x neurons per experiment * x trials * x shocks per trial. As such, the methodology here remains opaque, clouding the interpretation of the data.

Thanks for pointing this out. The legend of Figure 5B now reads: "Trial numbers are neuron numbers times three footshocks and footshock trials were sorted by their footshock response score (see method for details) from top to bottom. N = 179 and 211 cells were recorded from 7 and 6 of BLA^{Lypd1} and BLA^{Etv1} mice, respectively.